# Mobility-Embedded POIs: Learning What A Place Is and How It's Used from Human Movement

## Abstract

Recent progress in geospatial foundation models has highlighted the importance of learning general-purpose representations for real-world locations, particularly Points of Interest (POIs) where human activity concentrates. Yet, existing POI representations remain largely static, evolving from simple coordinates and metadata to visual features and, most recently, LLM-derived textual prompts, all of which describe what a place *is*, but not *how* it is actually used. We argue that human mobility provides a complementary and dynamic signal, capturing real-world visitation patterns that reveal how places function in practice. To this end, we introduce **Mobility Embedded POIs (ME-POIs)**, a pretraining framework that augments static text-embedding representations with mobility-derived signals from visit sequences, capturing dynamic usage patterns. Each visit is represented as a contextualized embedding that integrates the POI's static attributes with its temporal and sequential context, including when the visit occurs and which visits precede or follow it. To address the long tail of sparsely visited POIs, we transfer visit distributions from data-rich locations to sparse ones, leveraging multi-scale spatial proximity to capture local and regional patterns. We evaluate ME-POIs on large-scale human mobility datasets across a set of map enrichment tasks. We find that augmenting strong text embedding baselines with ME-POIs leads to consistent and substantial improvements across all tasks, confirming that mobility-informed embeddings offer complementary information that enhances static representations and enables a richer understanding of how places are used. Notably, even mobility embeddings alone, without any POI semantics, outperformed text-based embeddings on certain tasks, underscoring a key novelty of our approach.

## 1 Introduction

The increasing availability of large-scale geospatial data, together with advances in machine learning, has substantially advanced the analysis of urban and geographic environments (Lee & Kang, 2015; Bommasani et al., 2021). As the range of geospatial applications expands, a key challenge lies in learning general-purpose representations of fundamental geographic entities to support a diverse range of downstream tasks (Mai et al., 2024; Siampou et al., 2025a). Among these geographic entities, Points of Interest—places that people visit during their everyday life, such as coffee shops, gyms, and landmarks—are especially important: they serve as the core units of human activity and interaction within cities. Learning high-quality POI representations is thus fundamental for enabling a broad spectrum of geospatial applications, including digital mapping, navigation, transportation planning, urban analytics, and location-based recommendation systems (Choudhury et al., 2024).

Existing approaches to POI representation learning primarily focus on encoding static attributes from geographic coordinates (Mai et al., 2020; Rußwurm et al., 2023; Siampou et al., 2025b) to additional visual and textual information (Li et al., 2023; Yan & Lee, 2024; Vivanco Cepeda et al., 2023; Klemmer et al., 2025). In particular, recent methods leverage large language models (LLMs) to enrich POI representations, due to their ability to encode extensive geographic and semantic knowledge from massive internet-scale data (Li et al., 2024; Cheng et al., 2025). These approaches have demonstrated that with carefully designed prompts, often augmented with map data and contextual neighborhood information, LLMs can achieve improved downstream performance on static, place-

centric tasks, such as POI classification, population prediction, and home value estimation (Manvi et al., 2024). However, such language-based representations remain fundamentally limited and incomplete by their reliance on static, historical data. In reality, it is the dynamic patterns of human activity, such as how often, when, and within which visit sequences a place is visited (i.e., which places typically precede and follow it), that define its role within the urban environment. For example, two nearby POIs such as a grocery store and a convenience store may appear similar in textual descriptions, yet their visitation patterns differ: grocery stores attract longer visits in evenings and weekends, while convenience stores receive brief visits throughout the day. Such behavioral signals help differentiate between similar places and reveal functional characteristics that static data alone cannot capture. Much like a word derives meaning from its use in context, the significance of a place emerges from the mobility flows it attracts and how it is used (Musleh et al., 2022).

In parallel, although prior research has explored leveraging human mobility data to learn POI representations, these efforts are primarily targeted at mobility-centric tasks, like next-location prediction (Feng et al., 2017; Zhao et al., 2017; Shimizu et al., 2020; Wan et al., 2021; Lin et al., 2021). In these approaches, POI embeddings are optimized to capture short-term personal movement dynamics, modeling the sequential order in which places are visited. While effective for predicting mobility behaviors, they are not explicitly designed for, nor directly transferable to, place-centric tasks that require an understanding of long-term, aggregated patterns of place usage and function.

In this work, we address this gap by introducing **Mobility-Embedded POIs (ME-POIs)**: a framework that augments static POI representations derived by text embedding models, by directly integrating large-scale human mobility signals. Starting from visit sequences, our approach encodes each visit as a contextualized embedding that reflects the static attributes of the POI and its temporal context within mobility patterns. These visit-level embeddings are then aligned with a learnable POI embedding via contrastive learning, ensuring that each POI representation incorporates aggregated behavioral information over time and across users. To address the common challenge of data sparsity for rarely visited POIs (Xu et al., 2024), we propose a distribution transfer mechanism that propagates temporal usage patterns from close by, frequently visited POIs, across multiple spatial scales, to those with limited data. This multi-scale strategy allows to capture local and regional behavioral trends and yields high-quality POI embeddings even in the long tail of the visit distribution.

We evaluate ME-POIs on two large-scale, real-world mobility datasets across four map enrichment tasks: weekly opening hours, permanent closure detection, popularity and price level inference. The attributes in these tasks are often incomplete, outdated, or difficult to maintain at scale, making them a strong demonstration of the value of our mobility-informed representations. To our knowledge, this is the first systematic evaluation of POI embeddings on such tasks. Across all benchmarks, augmenting strong text-embedding baselines with ME-POIs yields consistent and substantial improvements, with gains of up to $16.2\%$ for opening hours, $6.5\%$ for permanent closures, $81.9\%$ for popularity, and $75.1\%$ for price level (in F1). These results highlight that a single embedding can support diverse downstream tasks, underscoring the versatility of ME-POIs and their value for enriching place representations. Remarkably, even ME-POIs alone, without explicit POI semantics, outperformed text-based embeddings in certain tasks, further emphasizing the novelty and robustness of our approach. In summary, our contributions are:

• We propose **Mobility-Embedded POIs (ME-POIs)**, a framework that augments static, text-based POI representations with longitudinal embeddings derived from large-scale human mobility data.

• We introduce a multi-scale distribution transfer mechanism that addresses mobility data sparsity by propagating temporal usage patterns from frequently visited POIs to sparsely visited ones.

• We conduct the first systematic evaluation of mobility-informed POI embeddings on a set of map enrichment tasks, demonstrating substantial improvements over strong text embedding baselines.

## 2 RELATED WORK

**Static POI Representation Learning.** Existing approaches to POI representation learning primarily rely on static attributes to encode the semantic and geographic relationships between places. Several methods focus on representing location and neighborhood structure using features like geographic coordinates, proximity to other places, and local connectivity (Yan et al., 2017; Mai et al., 2020; Rußwurm et al., 2023; Klemmer et al., 2023; Siampou et al., 2025b). To further enrich POI

representations, recent work incorporates additional context by integrating information derived from satellite, street-view, or remote sensing imagery, enabling models to capture environmental and physical characteristics of each place (Ayush et al., 2021; Vivanco Cepeda et al., 2023; Mai et al., 2023; Fuller et al., 2023; Balsebre et al., 2024; Klemmer et al., 2025). Text is another important modality for POI representation. Recent advances include (i) geospatial language models (Li et al., 2022; 2023; Yan & Lee, 2024) pretrained to improve language model performance on specialized spatial tasks, such as toponym recognition and geo-entity typing, by jointly encoding text and geographic information and (ii) approaches that extract geospatial knowledge directly from LLMs (Chen et al., 2023; Liu et al., 2024; Cheng et al., 2025). For example, GeoLLM (Manvi et al., 2024) designs spatially informed prompts to query LLMs for predicting place-specific properties (e.g., population, wealth, education) directly from language model outputs. While these methods form a strong foundation for static POI representation, they do not incorporate dynamic human mobility patterns, which provide complementary behavioral signals and can further enhance POI embeddings.

**Mobility-Informed POI Representation Learning.** Human mobility data has long been used to model movement dynamics between POIs. Many existing methods leverage sequences of POI visits or trajectories to learn POI embeddings, typically employing self-supervised objectives that capture patterns of co-visitation and transitions between places. Early approaches, such as POI2Vec (Feng et al., 2017), adapt word embedding techniques from natural language processing, treating sequences of POI visits analogously to sentences to capture spatial co-visitation patterns. Subsequent approaches jointly encode both spatial and temporal orderings to account for when and where places are visited (Zhao et al., 2017; Wan et al., 2021), while others leverage hierarchical structures among POIs to enhance representation granularity (Shimizu et al., 2020). CTLE (Lin et al., 2021) uses a masked modeling objective, randomly masking POIs and visit times in a sequence and training the model to predict the masked values, encouraging embeddings to capture the surrounding context. While these approaches are effective for modeling short-term movement dynamics, the resulting embeddings are typically conditional on local trajectory context and are not explicitly designed to capture stable, long-term patterns of place usage required for inferring static, place-centric attributes.

**Geospatial Foundation Models and Broader Impact.** Recent research has focused on developing geospatial foundation models (GeoFMs), general-purpose representation learning frameworks that aim to unify spatial, textual, visual, and mobility signals for broad transferability across geospatial tasks (Mai et al., 2024; Agarwal et al., 2024). However, existing efforts rarely incorporate mobility-derived behavioral patterns, due to the complexity and sparsity of large-scale mobility data (Choudhury et al., 2024). Our work complements recent GeoFM advances enriching static POI embeddings with real-world mobility signals and behavioral patterns, leading to richer transferable representations that improve map enrichment tasks, traditionally addressed with static data. Although our focus is on POIs, the same framework can extend to other geospatial objects, such as regions, road segments, and buildings, broadening its applicability within GeoFMs.

## 3 PROBLEM FORMULATION

Let $\mathcal{P} = \{p_1, \ldots, p_N\}$ denote the set of POIs within a geographic region, where each POI $p \in \mathcal{P}$ is associated with a location $x_p \in \mathbb{R}^2$ and textual metadata (e.g., name, category, description). Let, also, $\mathcal{S} = \{s_1, \ldots, s_K\}$ be a collection of visit sequences, where each sequence $s_k = (v_1, \ldots, v_{L_k})$ represents the temporally ordered visits of a user. Each visit is defined as $v_i = (x_i, t_i^a, t_i^d)$, where $x_i \in \mathbb{R}^2$ are the coordinates of the visited POI, and $t_i^a, t_i^d \in \mathbb{R}$ are the arrival and departure times.

**Objective.** Given a set of static POI embeddings $\{z_p^{\text{static}} \in \mathbb{R}^d : p \in \mathcal{P}\}$, derived from a pretrained text embedding model applied to POI metadata, and the set of visit sequences $\mathcal{S}$, our goal is to learn a mapping function $f : \mathbb{R}^d \times \mathcal{S} \to \mathbb{R}^d$ that produces a *mobility-embedded* POI representation $z_p^{\text{ME}} = f(z_p^{\text{static}}, \mathcal{S})$, for each $p \in \mathcal{P}$. Here, $z_p^{\text{ME}} \in \mathbb{R}^d$ integrates the static attributes of $p$ with the mobility context captured by longitudinal visitation dynamics.

## 4 METHODOLOGY

In this section, we present our framework for learning mobility-enriched POI embeddings, as depicted in Figure 1. Our approach consists of the following modules: (i) a transformer-based visit sequence encoder, (ii) a contrastive learning module for learning global POI representations, (iii) a

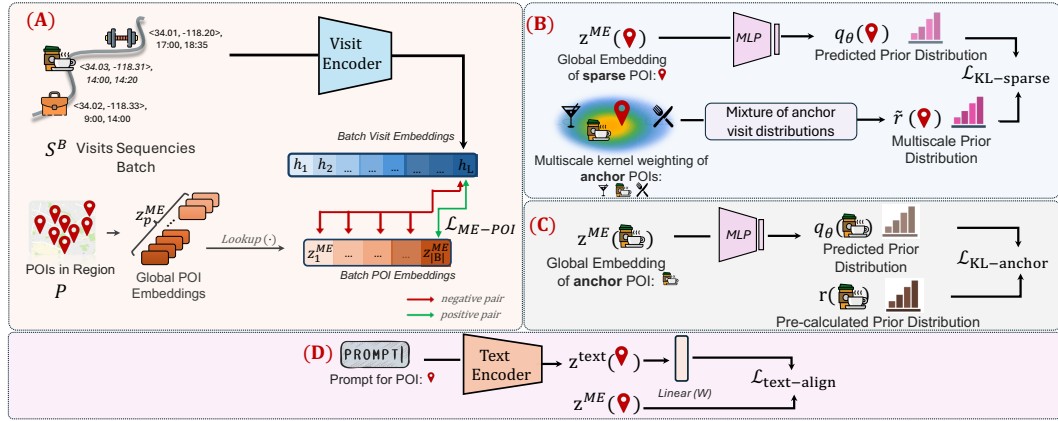

Figure 1: **Key components of ME-POIs pretraining:** (A) Contrastive learning aligns visit embeddings with their corresponding global POI embeddings. (B) Multi-scale priors transfer visit distributions from data-rich anchors to sparsely visised POIs. (C) An auxiliary loss aligns mobility embeddings with text embeddings for semantic grounding.

multiscale kernel-based distribution transfer module for sparse POIs, (iv) a direct supervision module for data-rich POIs to capture their temporal usage patterns, (v) and an auxiliary text alignment module to ensure compatibility with semantic text embeddings.

## 4.1 VISIT SEQUENCE ENCODER

We introduce a visit encoder model that operates on a batch of temporally ordered visit sequences $\mathcal{S}^B = \{s_1, \ldots, s_B\}$. For each sequence $s = (v_1, v_2, ..., v_L)$, the encoder outputs a sequence of contextualized visit embeddings $H = (h_1, h_2, ..., h_L)$, where $h_i$ captures both the local attributes of $v_i$ and its contextual role within the sequence.

**Visit Encoding.** Each visit $v_i$ comprises three main components: the geographical coordinates $x_i \in \mathbb{R}^2$ of the visited POI $p_i$, as well as its arrival and departure times $t_i^a, t_i^d \in \mathbb{R}$. We independently transform these components using three factorized encoders. Specifically, the location is embedded using a location encoder $\lambda_\theta : \mathbb{R}^2 \to \mathbb{R}^{d_l}$, while arrival and departure times are encoded via two separate time encoders $g_\eta, g_\zeta : \mathbb{R} \to \mathbb{R}^{d_t}$, reflecting their distinct semantic roles in characterizing each visit. In our implementation, we employ Theory Location Encoder (Mai et al., 2020) as $\lambda_\theta$, which provides a multiscale sinusoidal representation of coordinates[1], and Time2Vec (Kazemi et al., 2019) for $g_\eta$ and $g_\zeta$, to capture linear trends and periodic temporal patterns.

The resulting embeddings are then concatenated to form the initial visit encoding for $v_i$:

$$\tilde{h}_i^{(0)} = [\lambda_\theta(x_i) \,\|\, g_\eta(t_i^a) \,\|\, g_\zeta(t_i^d)] \in \mathbb{R}^{d_h}, \quad \text{where } d_h = d_l + 2d_t, \tag{1}$$

where $[\cdot \| \cdot]$ denotes vector concatenation.

**Transformer-based Sequence Modeling.** Given the sequence of visit embeddings $\tilde{H}^{(0)} = (\tilde{h}_1^{(0)}, \tilde{h}_2^{(0)}, \ldots, \tilde{h}_L^{(0)})$, our goal is to contextualize each visit by modeling its dependencies and interactions within the sequence. To achieve this, we employ a multi-layer Transformer encoder, which has become a standard architecture for capturing complex temporal and co-visitation patterns in trajectory modeling (Xue et al., 2021; Yang et al., 2022; Hsu et al., 2024; Xu et al., 2024).

To preserve temporal order, we first add a fixed sinusoidal positional encoding $\text{PE}(i) \in \mathbb{R}^{d_h}$ to each visit embedding, where $i$ denotes the index of the visit in the temporally sorted sequence. This yields a position-aware input representation:

$$h_i^{(0)} = \tilde{h}_i^{(0)} + \text{PE}(i) \tag{2}$$

---

[1]More advanced location encoders (e.g., Poly2Vec (Siampou et al., 2025b)) could be used when POIs are represented as richer spatial geometries (e.g., road segments as lines or building footprints as polygons)

The sequence of position-augmented embeddings $H^{(0)} = (h_1^{(0)}, h_2^{(0)}, \ldots, h_L^{(0)})$ is then processed by the Transformer encoder, which consists of stacked self-attention layers. Each Transformer layer comprises a multi-head self-attention module followed by a position-wise feedforward network (FFN), with residual connections and pre-layer normalization. Formally, a single layer computes:

$$H' = \text{LayerNorm}(H^{(0)} + \text{MultiHead}(H^{(0)})), \quad H^{(1)} = \text{LayerNorm}(H' + \text{FFN}(H')) \quad (3)$$

The multi-head attention mechanism is defined as:

$$\text{MultiHead}(H) = [\text{head}_1 \| \cdots \| \text{head}_j]W^O, \quad \text{head}_i = \text{Softmax}\left(\frac{HW_i^Q(HW_i^K)^\top}{\sqrt{d_k}}\right)HW_i^V, \quad (4)$$

where $W_i^Q, W_i^K, W_i^V \in \mathbb{R}^{d_h \times d_k}$ and $W^O \in \mathbb{R}^{jd_k \times d_h}$ are learnable projection matrices and $j$ is the number of heads.

Applying $N$ stacked Transformer layers yields the final contextualized visit embeddings:

$$H = (h_1, h_2, \ldots, h_L), \quad \text{where } h_i \in \mathbb{R}^{d_h} \text{ for } i = 1, \ldots, L \quad (5)$$

## 4.2 VISIT TO POI EMBEDDING CONTRASTIVE LEARNING

Given the individual contextualized visit vectors, we now describe how to learn global, usage-aware POI embeddings. To this end, we associate each POI $p \in \mathcal{P}$ with a global, learnable embedding vector $z_p^{\text{ME}} \in \mathbb{R}^{d_h}$, which is shared and updated across all occurrences of $p$ in the dataset. This embedding is designed to capture long-term, usage-aware semantics by aggregating behavioral information from every visit to $p$, thereby reflecting the full range of mobility patterns associated with that location. Unlike visit embeddings, which encode context-specific information for individual visits, $z_p^{\text{ME}}$ serves as a *unified representation that summarizes usage across all contexts*. One of the main novelties of our work lies in this departure from prior approaches that optimize POI embeddings primarily for sequential mobility prediction. Instead, we explicitly design embeddings that generalize to static, place-centric tasks requiring an understanding of long-term usage and function.

We achieve this aggregation through a contrastive learning framework. For each visit $v_i$ to POI $p$, we encourage the contextualized visit embedding $h_i$ to be similar to the global embedding $z_p^{\text{ME}}$, while dissimilar to embeddings of other POIs in the same batch. For this, we adopt the standard InfoNCE loss (Oord et al., 2018; Radford et al., 2021), which for a given visit $v_i$ to POI $p$ is defined as:

$$\mathcal{L}_{\text{ME-POI}}(h_i, z_p^{\text{ME}}) = -\log \frac{\exp(\text{sim}(h_i, z_p^{\text{ME}})/\tau)}{\sum\limits_{p' \in \mathcal{P}_{\text{batch}}} \exp(\text{sim}(h_i, z_{p'}^{\text{ME}})/\tau)}, \quad (6)$$

where $\text{sim}(a, b) = \frac{a^\top b}{\|a\|\|b\|}$ denotes cosine similarity and $\tau$ is a temperature hyperparameter.

This contrastive signal ensures that $z_p^{\text{ME}}$ is consistently updated toward visit embeddings associated with $p$, leading to a standalone representation that captures aggregated mobility patterns.

## 4.3 TRANSFERRING VISIT DISTRIBUTIONS TO SPARSE POIS

A common challenge in modeling human mobility is the long-tail distribution of visits across POIs: only a small subset of popular locations typically receives frequent visits, while the majority are sparsely visited by only a few users (Xu et al., 2024). This data imbalance can limit the ability of our contrastive framework to learn meaningful embeddings for underrepresented POIs, as these embeddings are updated with only a handful of visits. To address this, we introduce a visit distribution transfer mechanism that propagates temporal visitation patterns from frequently visited POIs (anchors) to sparsely visited ones, enabling reliable estimation of $z_{p_s}^{\text{ME}}$ even in low-data regimes.

We define a set of anchor POIs, $\mathcal{P}_{\text{anchor}} \subset \mathcal{P}$, as those with the highest total visit counts in the region of interest. For each anchor $p_a \in \mathcal{P}_{\text{anchor}}$, we compute an empirical weekly visit distribution $r_{p_a} \in \Delta^T$ by binning visits into $T$ fixed temporal slots (e.g., hourly intervals over a week) and normalizing the histogram to obtain a valid probability distribution.

To transfer these distributions, we leverage the empirical observation that geographically close POIs tend to exhibit similar visitation patterns (Miller, 2004). While semantic similarity (e.g., two restaurants) could, in principle, also reflect shared behavioral patterns (Zhu & Turner, 2022), our experiments showed that incorporating semantic features provided no improvement over using geographical distance alone. Moreover, these spatially-driven patterns appear at multiple resolutions, from local (block-level) similarities, such as neighboring coffee shops sharing morning peaks, to broader trends that distinguish neighborhoods or districts (e.g., residential versus commercial areas).

To capture this multiscale structure, we adopt a kernel-based approach that *combines distributions from anchors at varying spatial scales*, allowing each sparse POI to draw from both fine- and coarse-grained temporal signals. Specifically, we consider $M$ different spatial scales, each parameterized by a kernel bandwidth $\sigma_m$ for $m = 1, \ldots, M$. For each sparse POI $p_s \in \mathcal{P}_{\text{sparse}}$, we compute Gaussian kernel weights $\alpha$ over all anchors $p_a \in \mathcal{P}_{\text{anchor}}$ at each scale $\sigma_m$:

$$\alpha_{p_s,p_a}^{(m)} = \frac{\exp\left(-\frac{\|x_{p_s} - x_{p_a}\|^2}{2\sigma_m^2}\right)}{\sum_{p_a' \in \mathcal{P}_{\text{anchor}}} \exp\left(-\frac{\|x_{p_s} - x_{p_a'}\|^2}{2\sigma_m^2}\right)}, \tag{7}$$

where $x_{p_s}$ and $x_{p_a}$ denote the coordinates of the sparse POI and anchor, respectively.

We further learn mixture weights $\boldsymbol{\beta}_{p_s} \in \Delta^M$ for each sparse POI, which control the contribution of each spatial scale to the final distribution transfer. The resulting prior distribution is given by:

$$\tilde{r}_{p_s} = \sum_{m=1}^{M} \beta_{p_s,m} \left( \sum_{p_a \in \mathcal{P}_{\text{anchor}}} \alpha_{p_s,p_a}^{(m)} \cdot r_{p_a} \right) \tag{8}$$

To ensure that the learned embedding $z_{p_s}^{\text{ME}}$ encodes temporal usage patterns, we map $z_{p_s}^{\text{ME}}$ through a multi-layer perceptron (MLP) followed by a softmax to produce a predicted visit distribution:

$$q_\theta(p_s) = \text{softmax}(\text{MLP}(z_{p_s}^{\text{ME}})) \tag{9}$$

where $\text{MLP}(\cdot)$ denotes a neural network with one hidden layer and ReLU activation.

Finally, we train the model to align its predicted distribution $q_\theta(p_s)$ with the constructed prior $\tilde{r}_{p_s}$ using a KL divergence loss:

$$\mathcal{L}_{\text{KL-sparse}} = \sum_{p_s \in \mathcal{P}_{\text{sparse}}} \text{KL}\left(\tilde{r}_{p_s} \,\|\, q_\theta(p_s)\right) \tag{10}$$

### 4.4 DIRECT SUPERVISION FOR ANCHOR POIS

For anchor POIs with sufficient visit history, we directly supervise their embeddings to capture their observed temporal usage patterns. For each anchor POI $p_a \in \mathcal{P}_{\text{anchor}}$, we compute an empirical visit distribution $r_{p_a} \in \Delta^T$, and predict an approximate distribution $q_\theta(p_a) = \text{softmax}(\text{MLP}(z_{p_a}^{\text{ME}}))$ from the mobility embedding. Here, $\text{MLP}(\cdot)$ denotes the same network as for sparse POIs.

We then minimize the KL divergence between the empirical and predicted distributions:

$$\mathcal{L}_{\text{KL-anchor}} = \sum_{p_a \in \mathcal{P}_{\text{anchor}}} \text{KL}\left(r_{p_a} \,\|\, q_\theta(p_a)\right) \tag{11}$$

This loss complements the transfer loss for sparse POIs, ensuring that embeddings for anchors accurately reflect their observed visitation patterns.

### 4.5 ALIGNMENT WITH TEXT EMBEDDINGS

Our mobility-embedded POI representations are designed to *augment and complement static text embeddings for POIs*. For each POI, we derive a semantic embedding by passing a text prompt through a pretrained text embedding model. Following GeoLLM (Manvi et al., 2024), we construct a prompt for each POI, using POI information (i.e., coordinates, category, and address) and eighborhood context. We provide details related to the prompt construction in Appendix A.1.6. To

encourage the learned mobility embedding $z_p^{\text{ME}} \in \mathbb{R}^{d_h}$ to encode complementary semantic content, we project the text embedding into the mobility embedding space via a linear transformation $W \in \mathbb{R}^{d_h \times d_u}$. We then maximize the cosine similarity between $z_p^{\text{ME}}$ and the projected text embedding $W z_p^{\text{text}}$. Specifically, we use the following objective:

$$\mathcal{L}_{\text{text-align}} = \sum_{p \in \mathcal{P}} \left[ 1 - \cos\left( z_p^{\text{ME}}, \ W z_p^{\text{text}} \right) \right] \tag{12}$$

where $\cos(\cdot, \cdot)$ denotes cosine similarity.

### 4.6 MODEL OPTIMIZATION

**Pretraining.** The overall pretraining objective jointly optimizes four terms: (i) aligning contextualized visit representations with global POI embeddings via contrastive learning, (ii) regularizing anchor POI embeddings to match their empirical usage patterns, (iii) transferring temporal patterns to sparse POIs through KL supervision, and (iv) aligning mobility-based POI embeddings with semantic information from text embeddings. The total loss is:

$$\mathcal{L} = \mathcal{L}_{\text{ME-POI}} + \lambda_a \, \mathcal{L}_{\text{KL-anchor}} + \lambda_s \, \mathcal{L}_{\text{KL-sparse}} + \lambda_t \, \mathcal{L}_{\text{text-align}}, \tag{13}$$

where $\lambda_a$, $\lambda_s$, and $\lambda_t$ are hyperparameters controlling the contribution of each auxiliary loss term.

**Fine-Tuning.** For downstream evaluation, we freeze the pretrained embeddings and train only lightweight task-specific heads. Each POI $p$ is represented by two fixed vectors: the mobility-based embedding $z_p^{\text{ME}}$ and the text-based embedding $z_p^{\text{text}}$. To adapt these representations to a given task, we first project each through two separate small MLPs: $\tilde{z}_p^{\text{ME}} = \text{MLP}_p(z_p^{\text{ME}}), \tilde{z}_p^{\text{text}} = \text{MLP}_t(z_p^{\text{text}})$. We then concatenate the projected vectors and pass them to a task-specific prediction head:

$$\hat{y}_p = \text{MLP}_{\text{head}}\left( [\tilde{z}_p^{\text{ME}} \parallel \tilde{z}_p^{\text{text}}] \right) \tag{14}$$

Here, each MLP is a two-layer feedforward network with one hidden layer and ReLU activation.

## 5 EXPERIMENTS

**Datasets.** We use large-scale, anonymized human mobility datasets provided by Veraset[2], covering Los Angeles county and the city of Houston. The Los Angeles dataset spans a full calendar year, while the Houston dataset covers a 20-day period. Both datasets consist of raw GPS trajectories, containing timestamped geographic coordinates and randomized device identifiers. We convert the raw trajectories into sequences of visits by performing staypoint detection and POI attribution. We provide details on the algorithms in the Appendix A.1.2. POIs with at least $M$ visits are designated as anchors, while the remainder are considered sparse, with $M{=}100$ for Los Angeles and $M{=}50$ for Houston. Table 6 in Appendix A.1.1 summarizes the statistics of the datasets.

**Baselines.** We select a set of state-of-the-art text embedding models to serve as baselines for generating the static POI representations. Specifically, we consider **MPNET** (`all-mpnet-base-v2`) (Song et al., 2020), **E5** (`e5-large-v2`) (Wang et al., 2022), and **GTR-T5** (`gtr-t5-large`) (Ni et al., 2022) as widely used academic models, along with commercial embeddings from **Nomic** (`nomic-embed-text-v1`) (Nussbaum et al., 2024), **OpenAI** (`text-embedding-3-small`/`large`), and **Gemini** (`models/embedding-001`). For all baselines, we use the same POI prompts, as described in Section 4.5, to extract embeddings. To evaluate the performance of the static POI embeddings on the downstream tasks, we probe each model by training an MLP on the frozen text embeddings. We further select several widely used mobility-based POI embedding models originally developed for next-location prediction or sequential mobility modeling. These include **Skip-Gram** (Mikolov et al., 2013), **POI2Vec** (Feng et al., 2017), **Geo-Teaser** (Zhao et al., 2017), **TALE** (Wan et al., 2021), **HIER** (Shimizu et al., 2020), **CTLE** (Lin et al., 2021), **DeepMove** (Feng et al., 2018), **STAN** (Luo et al., 2021), **Graph-Flashback** (Rao et al., 2022), **GETNext** (Yang et al., 2022), and **TrajGPT** (Hsu et al., 2024). For a consistent comparison,

---
[2] https://www.veraset.com

we extract the POI embeddings each method produces after pretraining and evaluate them using the same frozen-embedding probing as the text baselines.

**Downstream Tasks.** We evaluate our approach on four map enrichment tasks: (i) multi-label classification of **weekly opening hours**, where the goal is to predict a 168-dimensional binary vector indicating the open/closed status of each POI for every hour of the week, (ii) binary classification of **permanent closure status**, (iii) ordinal classification of **popularity**, and (iv) ordinal classification of **price level**. Ground-truth labels for opening hours and permanent closures are obtained from SafeGraph[3], while popularity and price level are sourced from Google Maps by cross-referencing with SafeGraph POIs; both of them have four classes each from least to most popular and expensive, respectively. Note that the task of permanent closure status is excluded from the Houston dataset due to the absence of labels of sufficient quality. For each downstream task, we report two standard metrics appropriate to the prediction objective.

Table 1: Performance on map enrichment in Los Angeles. **Relative improvements highlighted**.

| Method | Open Hours F1 / AUROC | Permanent Closure F1 / AUPRC | Popularity Accuracy / F1 | Price Level Accuracy / F1 |
|---|---|---|---|---|
| ME-POIs (w/o $\mathcal{L}_{\text{text-align}}$) | $0.540_{0.002}$ / $0.703_{0.005}$ | $0.757_{0.025}$ / $0.154_{0.006}$ | $0.575_{0.004}$ / $0.257_{0.005}$ | $0.600_{0.008}$ / $0.308_{0.003}$ |
| MPNet | $0.542_{0.001}$ / $0.726_{0.004}$ | $0.736_{0.028}$ / $0.172_{0.005}$ | $0.600_{0.006}$ / $0.270_{0.006}$ | $0.615_{0.011}$ / $0.306_{0.007}$ |
| MPNet + **ME-POIs** | $0.628_{0.009}$ / $0.783_{0.007}$ | $0.766_{0.025}$ / $0.181_{0.003}$ | $0.610_{0.005}$ / $0.352_{0.003}$ | $0.662_{0.005}$ / $0.337_{0.003}$ |
| Improvement | 15.87% / 7.85% | 4.08% / 5.23% | 1.67% / 30.37% | 7.64% / 10.13% |
| E5 | $0.540_{0.001}$ / $0.722_{0.003}$ | $0.738_{0.031}$ / $0.176_{0.005}$ | $0.575_{0.005}$ / $0.184_{0.002}$ | $0.521_{0.021}$ / $0.189_{0.021}$ |
| E5 + **ME-POIs** | $0.601_{0.006}$ / $0.751_{0.003}$ | $0.786_{0.022}$ / $0.185_{0.004}$ | $0.602_{0.005}$ / $0.330_{0.005}$ | $0.632_{0.009}$ / $0.322_{0.004}$ |
| Improvement | 11.30% / 4.02% | 6.50% / 5.11% | 4.70% / 79.35% | 21.31% / 70.37% |
| GTR-T5 | $0.547_{0.001}$ / $0.721_{0.002}$ | $0.767_{0.018}$ / $0.173_{0.005}$ | $0.595_{0.004}$ / $0.241_{0.003}$ | $0.586_{0.026}$ / $0.278_{0.020}$ |
| GTR-T5 + **ME-POIs** | $0.618_{0.008}$ / $0.767_{0.004}$ | $0.774_{0.013}$ / $0.178_{0.006}$ | $0.615_{0.004}$ / $0.332_{0.001}$ | $0.654_{0.010}$ / $0.334_{0.004}$ |
| Improvement | 12.98% / 6.38% | 0.91% / 2.89% | 3.36% / 37.76% | 11.60% / 20.14% |
| Nomic | $0.539_{0.001}$ / $0.723_{0.003}$ | $0.749_{0.018}$ / $0.173_{0.009}$ | $0.586_{0.006}$ / $0.230_{0.004}$ | $0.614_{0.017}$ / $0.297_{0.013}$ |
| Nomic + **ME-POIs** | $0.619_{0.009}$ / $0.771_{0.006}$ | $0.762_{0.023}$ / $0.182_{0.006}$ | $0.603_{0.007}$ / $0.332_{0.003}$ | $0.659_{0.009}$ / $0.336_{0.005}$ |
| Improvement | 14.84% / 6.64% | 1.74% / 5.20% | 2.90% / 44.35% | 7.33% / 13.13% |
| OpenAI (small) | $0.547_{0.002}$ / $0.732_{0.002}$ | $0.695_{0.004}$ / $0.184_{0.008}$ | $0.599_{0.005}$ / $0.260_{0.004}$ | $0.637_{0.013}$ / $0.320_{0.007}$ |
| OpenAI (small) + **ME-POIs** | $0.632_{0.006}$ / $0.780_{0.005}$ | $0.696_{0.005}$ / $0.186_{0.006}$ | $0.617_{0.008}$ / $0.353_{0.010}$ | $0.675_{0.005}$ / $0.345_{0.003}$ |
| Improvement | 15.54% / 6.56% | 0.14% / 1.09% | 3.01% / 35.77% | 4.33% / 7.81% |
| OpenAI (large) | $0.548_{0.001}$ / $0.738_{0.004}$ | $0.750_{0.020}$ / $0.181_{0.006}$ | $0.607_{0.004}$ / $0.271_{0.003}$ | $0.654_{0.014}$ / $0.329_{0.007}$ |
| OpenAI (large) + **ME-POIs** | $0.637_{0.008}$ / $0.783_{0.005}$ | $0.770_{0.012}$ / $0.185_{0.007}$ | $0.626_{0.007}$ / $0.368_{0.004}$ | $0.684_{0.012}$ / $0.350_{0.006}$ |
| Improvement | 16.24% / 6.10% | 2.67% / 2.21% | 3.13% / 35.79% | 4.59% / 6.38% |
| Gemini | $0.548_{0.005}$ / $0.716_{0.006}$ | $0.756_{0.030}$ / $0.181_{0.006}$ | $0.581_{0.006}$ / $0.199_{0.005}$ | $0.559_{0.057}$ / $0.234_{0.059}$ |
| Gemini + **ME-POIs** | $0.613_{0.004}$ / $0.761_{0.004}$ | $0.753_{0.031}$ / $0.185_{0.006}$ | $0.614_{0.004}$ / $0.362_{0.004}$ | $0.672_{0.012}$ / $0.345_{0.008}$ |
| Improvement | 11.86% / 6.28% | -0.40% / 2.21% | 5.68% / 81.91% | 20.21% / 47.44% |

**Overall Results.** Table 1 and Table 2 report results for Los Angeles and Houston, respectively. Across both cities and all tasks, adding ME-POIs to any text embedding baseline yields consistent and often substantial gains. In Los Angeles, ME-POIs improve AUROC for open hours prediction by up to *7.85%*, and macro-F1 by up to *81.91%* for popularity and *70.37%* for price level prediction. Permanent closure detection also benefits, with AUPRC increasing by as much as *5.23%*. Results in Houston follow a similar trend: AUROC for open hours prediction improves by up to *8.66%*, while macro-F1 gains reach *61.57%* for popularity and *75.14%* for price level. The largest relative improvements occur in popularity and price level prediction tasks, where static text embeddings are limited. Text models can often recover such attributes for well-known places, where correlations are reinforced during pretraining, but they struggle for POIs in the long tail with sparse textual context. By injecting local visitation patterns, ME-POIs complement text embeddings and provide directly informative behavioral signals for these tasks.

We further evaluate a model variant trained exclusively on mobility objectives, which we term ME-POIs (w/o $\mathcal{L}_{\text{text-align}}$). This variant achieves competitive performance to text-based baselines, even surpassing them in certain tasks. For instance, in Los Angeles it outperforms E5 and MPNet on permanent closure detection, while in Houston it achieves higher price level prediction performance than GTR-T5 and Nomic. However, it does not consistently exceed the strongest text embeddings across all settings, likely due to its reliance on locally observed behavioral data: when the observation window is short, as in Houston with only 20 days of mobility traces, the learned representations lack sufficient behavioral diversity and coverage. By contrast, text embeddings always benefit from

---

[3]https://www.safegraph.com/

Table 2: Performance on map enrichment in Houston. **Relative improvements highlighted**.

| Method | Open Hours F1 / AUROC | Popularity Accuracy / F1 | Price Level Accuracy / F1 |
|---|---|---|---|
| ME-POIs (w/o $\mathcal{L}_{\text{text-align}}$) | $0.519_{0.003}$ / $0.604_{0.003}$ | $0.467_{0.007}$ / $0.263_{0.008}$ | $0.564_{0.013}$ / $0.276_{0.014}$ |
| MPNet | $0.653_{0.005}$ / $0.739_{0.005}$ | $0.539_{0.007}$ / $0.331_{0.011}$ | $0.599_{0.005}$ / $0.248_{0.004}$ |
| MPNet + **ME-POIs** | $0.725_{0.005}$ / $0.803_{0.002}$ | $0.548_{0.006}$ / $0.374_{0.005}$ | $0.687_{0.010}$ / $0.344_{0.006}$ |
| Improvement | 11.03% / 8.66% | 1.67% / 12.99% | 14.69% / 38.71% |
| E5 | $0.640_{0.011}$ / $0.754_{0.004}$ | $0.492_{0.007}$ / $0.229_{0.008}$ | $0.549_{0.008}$ / $0.177_{0.001}$ |
| E5 + **ME-POIs** | $0.690_{0.006}$ / $0.780_{0.002}$ | $0.538_{0.004}$ / $0.368_{0.003}$ | $0.635_{0.016}$ / $0.300_{0.009}$ |
| Improvement | 7.81% / 3.45% | 9.35% / 60.70% | 15.66% / 69.49% |
| GTR-T5 | $0.624_{0.005}$ / $0.742_{0.003}$ | $0.506_{0.006}$ / $0.257_{0.003}$ | $0.549_{0.008}$ / $0.177_{0.001}$ |
| GTR-T5 + **ME-POIs** | $0.713_{0.004}$ / $0.782_{0.002}$ | $0.544_{0.006}$ / $0.370_{0.004}$ | $0.645_{0.013}$ / $0.310_{0.009}$ |
| Improvement | 14.26% / 3.71% | 10.57% / 61.57% | 17.49% / 75.14% |
| Nomic | $0.721_{0.005}$ / $0.806_{0.004}$ | $0.504_{0.007}$ / $0.268_{0.007}$ | $0.578_{0.021}$ / $0.212_{0.019}$ |
| Nomic + **ME-POIs** | $0.738_{0.005}$ / $0.813_{0.003}$ | $0.538_{0.007}$ / $0.366_{0.005}$ | $0.667_{0.009}$ / $0.326_{0.007}$ |
| Improvement | 2.36% / 0.87% | 6.75% / 36.57% | 15.40% / 53.77% |
| OpenAI (small) | $0.654_{0.007}$ / $0.761_{0.004}$ | $0.537_{0.005}$ / $0.314_{0.010}$ | $0.595_{0.011}$ / $0.233_{0.008}$ |
| OpenAI (small) + **ME-POIs** | $0.743_{0.004}$ / $0.805_{0.003}$ | $0.569_{0.007}$ / $0.398_{0.004}$ | $0.729_{0.013}$ / $0.367_{0.007}$ |
| Improvement | 13.61% / 5.78% | 5.96% / 26.75% | 22.52% / 57.51% |
| OpenAI (large) | $0.702_{0.005}$ / $0.788_{0.004}$ | $0.552_{0.009}$ / $0.345_{0.007}$ | $0.601_{0.007}$ / $0.244_{0.004}$ |
| OpenAI (large) + **ME-POIs** | $0.761_{0.004}$ / $0.824_{0.002}$ | $0.578_{0.005}$ / $0.412_{0.005}$ | $0.758_{0.010}$ / $0.383_{0.005}$ |
| Improvement | 8.40% / 4.57% | 4.71% / 19.42% | 26.12% / 56.97% |
| Gemini | $0.676_{0.013}$ / $0.756_{0.004}$ | $0.521_{0.004}$ / $0.268_{0.002}$ | $0.549_{0.008}$ / $0.177_{0.001}$ |
| Gemini + **ME-POIs** | $0.741_{0.009}$ / $0.801_{0.002}$ | $0.565_{0.005}$ / $0.392_{0.006}$ | $0.634_{0.014}$ / $0.304_{0.012}$ |
| Improvement | 9.62% / 5.95% | 8.45% / 46.27% | 15.48% / 71.75% |

globally available corpora. Nevertheless, the best performance is always achieved when the two are combined, showing that mobility-derived representations provide unique, non-redundant information. Importantly, our experiments demonstrate that *a single embedding can support all four map enrichment tasks*, underscoring both the versatility of ME-POIs and their value for geospatial foundation models.

Table 3: Comparison with POI baselines on map enrichment in Los Angeles.

| Method | Open Hours F1 / AUROC | Permanent Closure F1 / AUPRC | Popularity AUROC / AUPRC | Price Level Accuracy / F1 |
|---|---|---|---|---|
| Skip-Gram | $0.462_{0.002}$ / $0.520_{0.006}$ | $0.649_{0.008}$ / $0.123_{0.004}$ | $0.530_{0.001}$ / $0.268_{0.001}$ | $0.564_{0.007}$ / $0.286_{0.004}$ |
| POI2Vec | $0.460_{0.003}$ / $0.482_{0.003}$ | $0.564_{0.039}$ / $0.112_{0.005}$ | $0.519_{0.003}$ / $0.263_{0.002}$ | $0.530_{0.014}$ / $0.249_{0.013}$ |
| Geo-Teaser | $0.460_{0.002}$ / $0.470_{0.005}$ | $0.448_{0.083}$ / $0.116_{0.004}$ | $0.523_{0.007}$ / $0.266_{0.003}$ | $0.511_{0.009}$ / $0.194_{0.023}$ |
| TALE | $0.461_{0.002}$ / $0.464_{0.006}$ | $0.375_{0.197}$ / $0.102_{0.003}$ | $0.486_{0.006}$ / $0.248_{0.003}$ | $0.504_{0.005}$ / $0.189_{0.027}$ |
| HIER | $0.473_{0.002}$ / $0.547_{0.004}$ | $0.660_{0.005}$ / $0.119_{0.001}$ | $0.569_{0.005}$ / $0.291_{0.001}$ | $0.529_{0.029}$ / $0.229_{0.047}$ |
| CTLE | $0.463_{0.001}$ / $0.511_{0.007}$ | $0.115_{0.102}$ / $0.098_{0.006}$ | $0.501_{0.006}$ / $0.249_{0.003}$ | $0.488_{0.015}$ / $0.244_{0.008}$ |
| DeepMove | $0.460_{0.003}$ / $0.484_{0.007}$ | $0.370_{0.135}$ / $0.110_{0.002}$ | $0.494_{0.006}$ / $0.253_{0.001}$ | $0.503_{0.009}$ / $0.224_{0.030}$ |
| STAN | $0.464_{0.002}$ / $0.509_{0.007}$ | $0.220_{0.215}$ / $0.099_{0.007}$ | $0.550_{0.006}$ / $0.250_{0.002}$ | $0.497_{0.012}$ / $0.248_{0.006}$ |
| Graph-Flashback | $0.463_{0.002}$ / $0.506_{0.008}$ | $0.233_{0.203}$ / $0.099_{0.007}$ | $0.504_{0.007}$ / $0.251_{0.002}$ | $0.496_{0.017}$ / $0.248_{0.009}$ |
| GETNext | $0.431_{0.007}$ / $0.500_{0.001}$ | $0.200_{0.220}$ / $0.103_{0.004}$ | $0.503_{0.001}$ / $0.252_{0.005}$ | $0.410_{0.092}$ / $0.220_{0.032}$ |
| TrajGPT | $0.483_{0.003}$ / $0.491_{0.005}$ | $0.215_{0.120}$ / $0.101_{0.006}$ | $0.496_{0.006}$ / $0.249_{0.003}$ | $0.475_{0.015}$ / $0.237_{0.009}$ |
| **ME-POIs** (w/o $L_{\text{text-align}}$) | $0.540_{0.002}$ / $0.703_{0.005}$ | $0.757_{0.025}$ / $0.154_{0.006}$ | $0.633_{0.004}$ / $0.337_{0.005}$ | $0.600_{0.011}$ / $0.308_{0.005}$ |
| **ME-POIs** | $0.554_{0.004}$ / $0.722_{0.005}$ | $0.766_{0.023}$ / $0.161_{0.005}$ | $0.653_{0.004}$ / $0.355_{0.008}$ | $0.609_{0.018}$ / $0.322_{0.012}$ |

**Comparison to mobility-informed POI representation baselines.** To highlight the benefits of our pretraining strategy for static, place-centric tasks, we also compare against widely used mobility-based POI embedding models originally designed for next-location prediction. We split these baselines into two categories: POI representation approaches (in the top portion of Tables 3 and 4) that learn dedicated POI embedding vectors as part of mobility-sequence objectives, and models (in the bottom portion) that provide POI embeddings implicitly via the learnable token-embedding layer of their next-location prediction architecture. For fairness, we report the performance of ME-POIs both with and without the text-alignment objective (w/o $L_{\text{text-align}}$), given that the baselines do not use any text signal. Notably, the mobility-only variant outperforms every mobility-based baseline across all tasks and in both cities. This result empirically supports our core hypothesis that next-location prediction models are optimized to capture short-term user transition dynamics, focusing on how individuals move from one place to another. These objectives do not encourage the model to learn the long-term, aggregated behavioral properties that characterize individual places. As a result, the POI embeddings they produce primarily encode sequential co-occurrence patterns rather

Table 4: Comparison with POI baselines on map enrichment in Houston.

| Method | Open Hours F1 / AUROC | Popularity AUROC / AUPRC | Price Level Accuracy / F1 |
|---|---|---|---|
| Skip-Gram | $0.483_{0.004}$ / $0.474_{0.005}$ | $0.558_{0.007}$ / $0.300_{0.004}$ | $0.543_{0.018}$ / $0.230_{0.013}$ |
| POI2Vec | $0.486_{0.004}$ / $0.503_{0.006}$ | $0.563_{0.006}$ / $0.298_{0.003}$ | $0.555_{0.027}$ / $0.270_{0.006}$ |
| Geo-Teaser | $0.483_{0.004}$ / $0.433_{0.002}$ | $0.504_{0.021}$ / $0.254_{0.012}$ | $0.514_{0.058}$ / $0.180_{0.025}$ |
| TALE | $0.482_{0.004}$ / $0.465_{0.004}$ | $0.507_{0.015}$ / $0.256_{0.007}$ | $0.529_{0.040}$ / $0.201_{0.028}$ |
| HIER | $0.498_{0.003}$ / $0.542_{0.009}$ | $0.519_{0.005}$ / $0.264_{0.001}$ | $0.551_{0.012}$ / $0.184_{0.006}$ |
| CTLE | $0.306_{0.013}$ / $0.496_{0.007}$ | $0.504_{0.009}$ / $0.258_{0.005}$ | $0.511_{0.012}$ / $0.230_{0.006}$ |
| DeepMove | $0.482_{0.004}$ / $0.454_{0.006}$ | $0.519_{0.009}$ / $0.262_{0.005}$ | $0.536_{0.021}$ / $0.230_{0.018}$ |
| STAN | $0.484_{0.004}$ / $0.496_{0.006}$ | $0.503_{0.009}$ / $0.257_{0.005}$ | $0.513_{0.012}$ / $0.231_{0.006}$ |
| Graph-Flashback | $0.484_{0.004}$ / $0.496_{0.007}$ | $0.505_{0.008}$ / $0.259_{0.005}$ | $0.510_{0.012}$ / $0.229_{0.005}$ |
| GETNext | $0.493_{0.003}$ / $0.551_{0.002}$ | $0.560_{0.004}$ / $0.293_{0.004}$ | $0.549_{0.013}$ / $0.180_{0.004}$ |
| TrajGPT | $0.483_{0.003}$ / $0.491_{0.006}$ | $0.501_{0.006}$ / $0.253_{0.004}$ | $0.534_{0.013}$ / $0.239_{0.008}$ |
| **ME-POIs** (w/o $\mathcal{L}_{\text{text-align}}$) | $0.519_{0.003}$ / $0.604_{0.003}$ | $0.570_{0.002}$ / $0.314_{0.004}$ | $0.564_{0.013}$ / $0.276_{0.014}$ |
| **ME-POIs** | $\mathbf{0.582_{0.007}}$ / $\mathbf{0.657_{0.006}}$ | $\mathbf{0.598_{0.004}}$ / $\mathbf{0.352_{0.004}}$ | $\mathbf{0.590_{0.010}}$ / $\mathbf{0.294_{0.011}}$ |

than long-term temporal visitation patterns or functional roles. This mismatch leads to consistently weaker performance on map-enrichment tasks. Finally, our full ME-POIs model, which incorporates the text-alignment objective, achieves the strongest overall performance. This version reflects the intended use of the framework, where mobility-derived behavioral signals enrich and strengthen semantic POI representations.

**Ablation Study.** Table 8 presents the incremental contribution of each component in our framework. Starting from the base contrastive loss ($\mathcal{L}_{\text{ME-POI}}$), adding $\mathcal{L}_{\text{KL-sparse}}$ further improves results by regularizing long-tail POIs with anchor-derived visitation priors. This is especially evident in Los Angeles, where anchor coverage is denser. Adding $\mathcal{L}_{\text{KL-anchor}}$ yields additional but moderate gains, as anchors represent only a small subset of POIs. Finally, incorporating $\mathcal{L}_{\text{text-align}}$ loss, further enhances performance

Table 5: Ablation on ME-POIs for open hours prediction.

| Method | Los Angeles F1 / AUROC | Houston F1 / AUROC |
|---|---|---|
| ME-POIs ($L_{\text{ME-POI}}$) | $0.490_{0.004}$ / $0.608_{0.004}$ | $0.510_{0.004}$ / $0.595_{0.005}$ |
| + $L_{\text{sparse}}$ | $0.535_{0.005}$ / $0.701_{0.005}$ | $0.518_{0.004}$ / $0.603_{0.005}$ |
| + $L_{\text{anchor}}$ | $0.540_{0.002}$ / $0.703_{0.005}$ | $0.519_{0.003}$ / $0.604_{0.003}$ |
| + $L_{\text{text-align}}$ | $\mathbf{0.554_{0.004}}$ / $\mathbf{0.722_{0.005}}$ | $\mathbf{0.582_{0.007}}$ / $\mathbf{0.657_{0.006}}$ |

by grounding mobility-derived embeddings in semantic context. Here, results are obtained by aligning with OpenAI-large text embeddings. Overall, each objective provides complementary benefits, and the full combination achieves the strongest results.

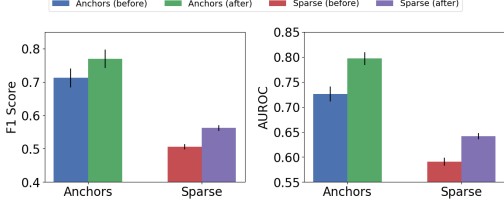

Figure 2: Effect of distribution transfer for Houston open-hours prediction.

**Case Study (Impact of distribution transfer).** To evaluate the benefit of the distribution transfer module, we report downstream performance for anchor POIs and sparse POIs before and after applying $\mathcal{L}_{\text{KL-anchor}}$ and $\mathcal{L}_{\text{KL-sparse}}$. As shown in Figure 2, distribution transfer consistently improves F1 and AUROC for both groups. This indicates that sparse POIs benefit from the multiscale temporal transfer, while anchor POIs improve through direct KL supervision. We further observe that anchors achieve higher absolute performance, as expected given their stronger mobility signal. Overall, these results confirm that the proposed distribution transfer module improves representation quality for both groups.

## 6 CONCLUSION

We proposed ME-POIs, a pretraining framework that augments static text embedding representations with mobility-derived signals from visit sequences, effectively capturing dynamic usage patterns. Our experiments demonstrate that adding ME-POIs to strong text embedding baselines yields consistent and substantial improvements across all tasks, confirming that mobility-informed embeddings provide complementary information and enable a richer understanding of how places are used. Future work will extend our framework to represent other geospatial objects, including road segments, administrative boundaries, and regions. This underscores that the impact of our work extends beyond POI embeddings to a wider spectrum of geospatial representations.

REPRODUCIBILITY STATEMENT

We have taken several steps to ensure the reproducibility of our work. The codebase implementing our models, training and evaluation pipelines will be released publicly upon acceptance. To facilitate replication, we provide detailed descriptions of all model architectures, training objectives, and optimization settings in the main paper, and report the exact hyperparameters used in our experiments in the Appendix A.1.3. Our experiments are conducted primarily on large-scale human mobility datasets from Veraset and POI data from SafeGraph, which can be accessed by researchers upon request. We also describe the dataset preprocessing steps we followed, including the algorithms used for staypoint detection and visit attribution, in Appendix A.1.2. Together, these resources enable researchers to replicate our results and build upon our work.

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

# A  APPENDIX

## A.1  ADDITIONAL DETAILS ON EXPERIMENTAL SETUP

### A.1.1  DATASET STATISTICS

We present the dataset statistics on Table 6. The number of POIs for both urban areas are comparable (LA has a larger bounding box and hence more PoIs). However, the number of visits for LA is an order of magnitude larger due to the year-long time-span, compared to 20 days for Houston.

Table 6: Summary of dataset statistics.

| Region | Time Period | Bounding Box | # POIs | # Visits | % Anchor POIs |
|---|---|---|---|---|---|
| Los Angeles | 01/01 - 12/31 2019 | [32.81, -118.94, 34.82, -117.65] | 39,557 | 6,908,365 | 9.07% |
| Houston | 03/05 - 03/26 2020 | [29.55, -95.56, 29.95, -95.16] | 28,419 | 715,604 | 7.04% |

### A.1.2  DATASET PREPROCESSING

We perform staypoint detection and POI attribution to convert our initial raw GPS trajectories into sequences of visits. For staypoint detection, we use the `trackintel` library, which implements the standard distance-time threshold method proposed by Li et al. (2008), designating a stay whenever the user remains within a *dist_threshold*=100 m radius for at *time_threshold*=5 minutes. For POI attribution, using POI geometries and locations from SafeGraph, we assign each visit to a POI if its location falls inside the POI's polygon, or otherwise to the nearest centroid within 100 meters. Visits that cannot be matched are labeled as UNKNOWN. These visits are kept in the sequences to preserve the temporal continuity of user trajectories but are excluded from the loss computation since they lack reliable POI labels. After preprocessing, we exclude sequences with less than 5 visits, to ensure sufficient context.

### A.1.3  IMPLEMENTATION DETAILS & HYPERPARAMETER CONFIGURATION

We normalize all coordinates to the range $[0, 1]$ using the bounding box of each area of interest. For the Theory Location Encoder, we set $\lambda_{\max} = 1.4142$ (the normalized diagonal distance), $\lambda_{\min} = 0.1$, and use 64 scales. Temporal features are normalized to $[0, 1]$ by extracting the hour within the day and the day within the week. Each is encoded separately and then combined into a single temporal representation. For the Gaussian kernels, we use scales of 0.3, 1.0 and 3.0 km, which are subsequently normalized to match the coordinate normalization.

Model hyperparameters are set as follows: sequence window size $w$=32, embedding dimension $d_h$=512, text embedding dimension $d_u$=768, number of attention heads $i$=8, feedforward hidden size 1024, and $N$=4 Transformer layers. All MLPs consist of a single hidden layer with dimension 256 and ReLU activation. We pretrain the model on the entire visit sequence dataset, and then fine-tune with a 60/20/20 train/validation/test split. We use Adafactor optimizer for pretraining with learning rate $1e - 3$ and AdamW during fine-tuning, with learning rate $1e - 5$. We pretrain the

model for 20 epochs and finetune it for up to 100 epochs with early stopping. Lastly, we set the hyperpaarameters $\lambda_\alpha = \lambda_s = \lambda_t = 1$.

### A.1.4 EXPERIMENTAL ENVIRONMENT

We implement our models in PyTorch 2.6.0 on a Debian Linux server, equipped with 50 GB RAM, 8 vCPUs (Intel Xeon @ 2.30 GHz), and an NVIDIA Tesla V100–SXM2–16GB GPU (CUDA 13.0).

### A.1.5 DOWNSTREAM TASKS & LABELS

We evaluate our approach across four downstream tasks: (i) open hours prediction, (ii) permanent closure detection, (iii) venue popularity classification, and (iv) price level classification. For each task, we keep only POIs with available labels, so the counts differ across tasks. In Los Angeles, 16,692 POIs have open hours labels, while in Houston, 14,465 POIs have open hours labels. For permanent closure, we assume that POIs with missing labels are not permanently closed; under this assumption, 3,807 POIs in the Los Angeles dataset are labeled as permanently closed. For popularity, 22,369 POIs in Los Angeles and 15,632 POIs in Houston have available labels. For price level, 5,091 POIs in Los Angeles and 4,105 POIs in Houston have available labels. Per-label statistics for the popularity and price level tasks are reported in Table 7.

Table 7: Venue Popularity and Price Level Counts

| Class | Los Angeles | | Houston | |
|---|---|---|---|---|
| | Popularity | Price Level | Popularity | Price Level |
| 0 | 12840 | 2563 | 7158 | 2270 |
| 1 | 1376 | 2311 | 979 | 1675 |
| 2 | 5654 | 181 | 4841 | 133 |
| 3 | 2499 | 36 | 2654 | 27 |

### A.1.6 TEXT EMBEDDING MODELS AND PROMPTS

We construct text prompts for each POI following the GeoLLM (Manvi et al., 2024) approach, which incorporates both (i) POI information, including coordinates, category, and address, which we obtain from Safegraph and (ii) neighborhood context, including the name, distance, and direction of the 10 closest POIs. This prompt design has been shown to effectively extracts geospatial knowledge, producing text embeddings that captures rich semantic and contextual information. We then query text embedding models (e.g., OpenAI and Gemini), and set the output dimension to 768, to ensure a fair comparison across models.

An example prompt is shown below:

> **Taco Man** (Restaurants and Other Eating Places). Coordinates: 34.062307, -118.197612. Address: 1602 N Soto St, Los Angeles, CA, 90033.
>
> **Nearby Places:**
> 0.0 km West: Tacos La Guera;
> 0.0 km West-Southwest: Soto Liquor Market;
> 0.1 km West: DaVita;
> 0.1 km West: Davita Trc Usc Kidney Center;
> 0.2 km North-Northeast: Ai Food Corporation;
> 0.2 km West: USC Occupational Therapy Faculty Practice;
> 0.2 km West: Molecular Imaging Center;
> 0.2 km West-Southwest: Bright Horizons Usc Hsc Infant Care Center;
> 0.2 km West-Southwest: Bright Horizons Usc Hsc Child Development Ctr;
> 0.3 km Northeast: Cardinal Moving Systems.

Figure 3: Example prompt for Taco Man POI in Los Angeles.

## A.2 ADDITIONAL EXPERIMENTS

### A.2.1 EMBEDDING VISUALIZATION

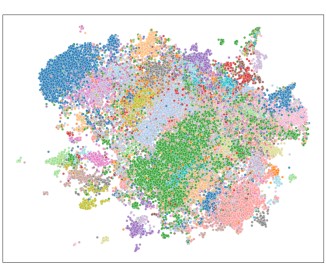 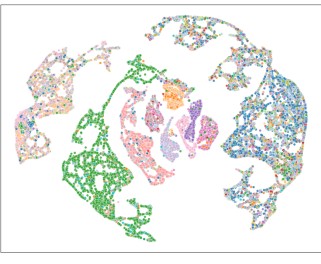 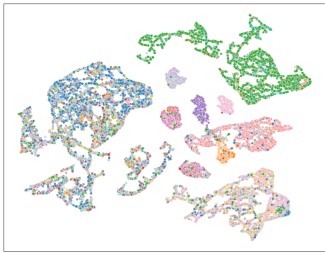

    (a) Text embeddings       (b) ME-POIs (contrastive only)     (c) ME-POIs (w KL transfer)

Figure 4: **UMAP visualization of POI embeddings in Los Angeles, colored by SafeGraph top category (141 classes).** No category information is provided to the models during pretraining. (a) Text embeddings form an unstructured cloud with limited category separation. (b) Mobility-based contrastive embeddings exhibit stronger clustering by functional category. (c) Adding KL-based transfer further sharpens the separation between categories, despite no category supervision.

To qualitatively evaluate the structure captured by our learned representations, we visualize the POI embeddings for Los Angeles using UMAP, coloring each point by its SafeGraph top category (141 unique classes). Importantly, no such category information was used during pretraining ME-POIs. To that extent, we compare three variants: (i) Text embeddings, generated by OpenAI text embedding model (`text-embedding-3-large`) using our prompts, (ii) ME-POIs (contrastive only) trained only with our contrastive learning objective ($\mathcal{L}_{\text{ME-POI}}$), and (iii) ME-POIs (w KL transfer), including the KL transfer objectives.

As shown in Figure 4, the text embeddings yield an unstructured, cloud-like distribution, with only broad clusters for the most common categories. In contrast, our mobility-based embeddings exhibit much stronger organization by functional category, *even though category information is never provided to the model*. Notably, after introducing KL-based distribution transfer, the clusters corresponding to major categories become even more well-defined, with boundaries that align closely with ground-truth POI types. These results demonstrate that mobility-derived representations naturally recover functional and behavioral groupings among places, offering complementary information to text models. The clear emergence of category structure, without any supervision, highlights the expressiveness and generality of our approach for place representation.

### A.2.2 CASE STUDY

To illustrate the benefits of mobility-based POI embeddings, we examine two nearby retail stores in Los Angeles: Circle K (a 24-hour grocery store) and Domaine LA (a wine store). Both are within 0.0021 degrees of each other and share similar SafeGraph retail categories, making them nearly indistinguishable in terms of text and neighborhood context.

Despite this, their temporal and behavioral patterns for these places differ substantially. Circle K is open 24/7 and attracts short, spotaneous visits throughout the week, while Domaine LA operates only during limited afternoon and evening hours, serving a more specialized customer base. As shown in Figure 5, the two POIs are mapped closely together in the text embedding space, but are clearly separated in the mobility embedding space. This separation reflects their distinct operational and visitation patterns, which are not captured by static attributes. This case study highlights how mobility-derived embeddings reveal behavioral differences among POIs that appear similar in text.

### A.2.3 COMPARISON WITH MASKED LANGUAGE MODELING

To show the value of our contrastive objective, we compare against masked language modeling (MLM), a widely used self-supervised objective in mobility representation learning. Specifically, we adapt the pretraining objectives of CTLE (Lin et al., 2021) as a representative variant, which enables a direct comparison between masked modeling and our contrastive objective. CTLE is

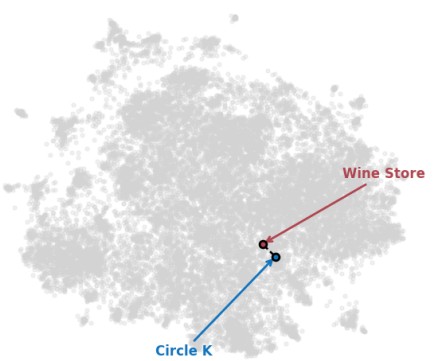 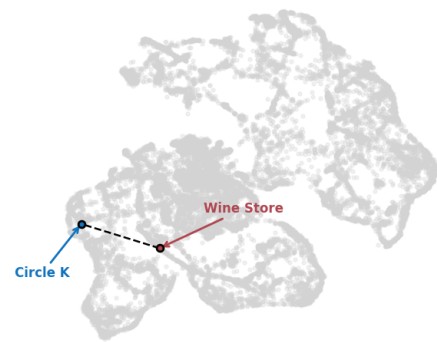

(a) Text embedding UMAP: POIs with similar descriptions are co-located.

(b) Mobility embedding UMAP: POIs with different usage are separated.

Figure 5: **Case study: Comparing two semantically similar and close by places (*Circle K* and *Domaine LA Wine Store*) in Los Angeles.** (a) In text embedding space, the POIs are nearly indistinguishable. (b) In mobility embedding space, they are separated, reflecting their different visitation patterns.

currently the state-of-the-art model for POI representation Cheng et al. (2025). We evaluate two variants of this baseline, where we randomly mask $25\%$ of the visits in each sequence:

- **MLM-POI:** We mask POI identifiers within a sequence and train the model to predict the masked POI from its surrounding context. This objective encourages embeddings to capture co-visitation and local transition patterns.

- **MLM-POI+Time:** In addition to masking POI identifiers, we also mask arrival and departure times. The model jointly predicts the masked POI and its temporal attributes (discretized into time bins), encouraging embeddings to capture both spatial and temporal context.

Table 8: Comparison of MLM baselines and ME-POIs for open hours and permanent closure prediction in Los Angeles. **Best** values are highlighted.

| Method | Open Hours
F1 / AUROC | Permanent Closure
F1 / AUPRC |
|---|---|---|
| MLM-POI | $0.461_{0.002}$ / $0.474_{0.006}$ | $0.319_{0.009}$ / $0.102_{0.005}$ |
| MLM-POI + Time | $0.461_{0.002}$ / $0.482_{0.005}$ | $0.402_{0.120}$ / $0.103_{0.005}$ |
| ME-POIs ($L_{\text{ME-POI}}$) | $0.490_{0.004}$ / $0.608_{0.004}$ | $0.755_{0.021}$ / $0.155_{0.005}$ |

The results in Table 8 show a clear gap between MLM and our contrastive formulation. We believe this is because MLM-POI captures short-range co-visitation patterns but remains limited to predicting masked elements within a single trajectory window, making the resulting embeddings highly context-dependent. Incorporating temporal attributes in MLM-POI+Time provides a modest boost, since visit timing does carry useful behavioral information, but the improvement is small because the objective is still confined to local sequence recovery. In contrast, ME-POIs substantially outperforms both MLM variants because it is not restricted by a context window. By aligning all visit representations with a single POI embedding, the contrastive loss aggregates information across sequences and users, producing embeddings that reflect long-term usage patterns. This design makes ME-POIs much more effective for static, place-centric tasks such as open hours and closure prediction.

### A.3 ENCODING

#### A.3.1 LOCATION ENCODING

The location encoder $\lambda_\theta$ is based on the Theory Location Encoder (Mai et al., 2020), which maps $x \in \mathbb{R}^2$ into a multi-scale sinusoidal representation. Specifically, we project $x$ onto three fixed directions $a \in \mathbb{R}^2$, and for each scale $s = 0, \ldots, S-1$ compute

$$\text{PE}(x; a, s) = \left[ \cos\left( \frac{\langle x, a \rangle}{\lambda_{\min} g^{s/(S-1)}} \right), \ \sin\left( \frac{\langle x, a \rangle}{\lambda_{\min} g^{s/(S-1)}} \right) \right], \tag{15}$$

where $g = \lambda_{\max}/\lambda_{\min}$. Concatenating all $3S$ such pairs yields a $6S$-dimensional vector, which is passed through an MLP to produce the final location embedding $\lambda_\theta(x) \in \mathbb{R}^{d_l}$.

#### A.3.2 TIME ENCODING

The time encoders $g_\eta, g_\zeta$ are implemented following Time2Vec (Kazemi et al., 2019), which maps a scalar input $t \in \mathbb{R}$ to a $d_t$-dimensional embedding:

$$g(t) = \left[ \omega_0 t + \phi_0, \ \sin(\omega_1 t + \phi_1), \ldots, \sin(\omega_{d_t-1} t + \phi_{d_t-1}) \right], \tag{16}$$

where $\omega_i, \phi_i$ are learnable parameters. The first component captures linear trends, while the remaining components capture periodic temporal patterns.

### A.4 COMPUTATIONAL EFFICIENCY

The pre-training cost of ME-POIs is dominated by running the visit encoder on sequences of visits. For a sequence length of $L$ and an embedding dimension $d$, the overall computation complexity is $O(L^2 \cdot d + L \cdot d^2)$ for. The contrastive module operates only over in-batch negatives: for a batch of $B$ visits containing $U$ unique POIs, its cost is $O(B \cdot U \cdot d)$, which in practice remains lightweight and independent of the full POI set size. Note that the # of unique POIs in the batch is less than or equal to # of visits in the batch. The POI anchor distributions and multiscale kernels are precomputed only once offline, with computation complexity $O(M \cdot |\mathcal{P}_{\text{anchor}}| \cdot |\mathcal{P}_{\text{sparse}}|)$ for $M$ scales. In practice, our model is lightweight with 53.7 M parameters, well within standard computational budgets.

### A.5 THE USE OF LARGE LANGUAGE MODELS (LLMS)

We used large language models (LLMs) during the preparation of this paper exclusively to polish the writing and to assist with figure visualization scripts. All research contributions, including ideation, model development, theoretical analysis, and experimental evaluation, were conducted entirely by the authors.

