# OpenReview forum: "Mobility-Embedded POIs: Learning What a Place Is and How It’s Used from Human Movement"
_ICLR.cc/2026/Conference — Submitted to ICLR 2026_

### Official Review · Reviewer_Fo4i · 2025-10-24

**Soundness:** 3
**Presentation:** 2
**Contribution:** 1
**Rating:** 2
**Confidence:** 4

**Summary:**

The paper focuses on learning general representation of POIs and introduces a new embedding method leveraging information from human mobility data. The proposed method can be used with map enrichment methods and experiments demonstrate its effectiveness.

**Strengths:**

1. The presentation of challenges and the motivation are detailed and intuitive.
2. A general embedding method is proposed demonstrating effectiveness when combined with several methods for the map enrichment task.

**Weaknesses:**

1. As also pointed out in the paper itself, there are several existing methods that have explored the idea of extracting POI information from human mobility data. It seems that the core idea is mostly shared between the proposed method and these existing ones, and the uniqueness and technical contribution of the proposed method on top of the core idea is not very obvious.
2. The experimental setting can be expanded to further demonstrate the generalizability of the proposed embedding method. Currently it is not tested on other common tasks related to POIs and human mobility data, such as POI recommendation (next location prediction), POI visiting flow prediction, etc.

**Questions:**

Could the authors elaborate on how their proposed method advances on top of the core idea of extracting POI information from human mobility data that is shared by many existing methods?

---

> ### Author Response · Authors · 2025-11-21
>
> **Comment W1 & W3:** *“It seems that the core idea is mostly shared between the proposed method and these existing ones, and the uniqueness and technical contribution of the proposed method on top of the core idea is not very obvious.”*
>
> **Response:** Prior methods that incorporate mobility signals for POIs primarily focus on local, sequence-level transitions (e.g., next-POI prediction), where Transformer approaches are used as analogues of next-word models. In contrast, our framework addresses a different and previously unexplored problem: learning global, usage-aware POI representations that capture how places are visited across many users, times, and contexts, and can serve as strong complementary embeddings to text-derived POI representations to address static POI tasks.
>
> Because this problem requires modeling aggregated behavioral patterns, our framework begins with the standard use of the Transformer encoder architecture for visit encoding but introduces two components not found in prior work: (i) a global contrastive learning objective that aligns each contextualized visit embedding with a shared, learnable POI embedding, to accumulate population-level behavioral information from all its visits; and (ii) a multiscale distribution transfer mechanism that provides temporal visitation priors for sparsely visited POIs. To our knowledge, neither of these components has been explored in prior work. In particular, we are not aware of any method that uses contrastive learning to aggregate long-term usage patterns (mobility-based or otherwise) into global POI embeddings. Likewise, no work has demonstrated that such mobility signals can meaningfully improve text-based POI representations derived from modern text encoders.
>
> We have additionally included new experiments  (please see Table 3&4 in Section 5) showing that POI embeddings derived from sequence-level mobility models used in next-location prediction tasks perform significantly worse on our static downstream tasks compared to embeddings built from our ME-POIs, which is expected given that the objectives of these next-location prediction models differ fundamentally from those of general-purpose POI representation learning.
>
> In line with the scope of ICLR, our work presents **a new representation-learning formulation for POIs**, and is the first modeling framework that implements this idea. We believe that this conceptual shift expands the field and creates space for future research to incorporate more sophisticated architectures within this formulation.
>
> **Comment W2:** *“Currently the approach is not tested on other common tasks related to POIs and human mobility data, such as POI recommendation (next location prediction), POI visiting flow prediction, etc.”*
>
> **Response:** We would like to clarify that tasks such as next-location prediction or flow prediction are not aligned with the purpose of ME-POIs. These tasks focus on predicting user movement sequences, while ME-POIs is specifically designed to learn global POI representations that capture long-term behavioral patterns for static map-enrichment tasks. As such, evaluating on sequential mobility-prediction tasks would not be directly meaningful for assessing our approach in its intended setting.

---

> ### Comment · Reviewer_Fo4i · 2025-11-26
>
> Thank you for the detailed reply.
>
> From my understanding, the focus of the proposed POI embedding is on learning a global representation shared across different users and temporal context.
> To some extent, all previous works that learn one embedding vector for each POI are learning a global representation for that POI. Regardless of the detailed embedding learning method, one can say that the embedding vector is learned on the "global" dataset, inevitably combining global information to some extent.
> Thus, I still believe that the claim that previous works only incorporate local, sequence-level transitions is not accurate.
> Some works even explicitly incorporate global information into their learned POI representations. For example, TALE [1] incorporates temporal visiting patterns to POIs across users, and POI2Vec [2] incorporates spatial correlations between POIs. Both of which are global information.
>
> Seeing that the fundamental problem definition is not completely new, the technical contribution of the proposed solution is a bit lacking.
> Although there might not exist a POI representation learning work that uses exactly the same technical framework as the proposed method, it appears that most of the techniques used in the proposed method are adopted, well-established ones.
>
> Regarding experiments on other POI-related downstream tasks, I still believe that given the fundamental problem of learning POI representations, it would be nice if the learned embeddings can be evaluated on more tasks.
> It shouldn't be a conflicting focus that a set of learned POI representations can be applied to both local-focused and global-focused downstream tasks, for example, POI2Vec [2] is evaluated on both next POI prediction (local-focused) and POI visiting flow prediction (global-focused) tasks.
> Nevertheless, the newly added experiments of evaluating prior works on the global-focused task do partially improve the comprehensiveness of the experiments.
>
> Anyway, these are my current reflections. The authors are welcome to correct me or elaborate on the above points further.
>
>
> [1] Wan, Huaiyu, et al. "Pre-training time-aware location embeddings from spatial-temporal trajectories." IEEE Transactions on Knowledge and Data Engineering 34.11 (2021): 5510-5523.
>
> [2] Feng, Shanshan, et al. "Poi2vec: Geographical latent representation for predicting future visitors." Proceedings of the AAAI Conference on Artificial Intelligence. Vol. 31. No. 1. 2017.

---

> > ### Author Response · Authors · 2025-11-29
> >
> > We agree with the reviewer that prior methods such as POI2Vec and TALE learn one embedding per POI and that they inevitably incorporate dataset-level information. We do not claim that these approaches use “local data” in a literal sense. Our point is that prior methods differ fundamentally in their optimization objectives and supervision signals. Some might incorporate temporal patterns, spatial correlations or POI information etc, however, the embeddings are ultimately optimized through mobility-centric objectives (next-location prediction). As a result, POIs are learned indirectly through their role in sequences, rather than being explicitly trained to encode long-term, place-level behavioral properties. Our work takes a different position by defining POI representation learning as a place-centric problem, and not a byproduct of sequence modeling. This shift between sequence-centric learning and place-centric learning is what we refer to when discussing “local” versus “global” information.
> >
> > Our newly added results further highlight this distinction. Specifically, models like POI2Vec and TALE that the reviewer mentions significantly underperform compared to our proposed framework. We believe this confirms that objectives designed for mobility prediction do not transfer well to map enrichment tasks. With that in mind, we disagree with the reviewer that our problem formulation is not fundamentally new.  In fact, next-POI prediction has been an active research area for many years, yet no prior work has considered using the resulting embeddings as a signal for understanding how a POI is used in a place-centric downstream task.
> >
> > Regarding our technical contributions, we agree that contrastive learning is a well-known mechanism. However, unlike prior work that applies contrastive learning between augmented views or across modalities, we use it to directly align visit-level representations with their corresponding POI embeddings, so that a single embedding per place is learned from its visits. To the best of our knowledge, this formulation has not been previously explored in the domain.
> >
> > Finally, we emphasize that the primary goal of this work is to demonstrate that human mobility is a missing but highly informative signal for improving modern text embeddings of places. We introduce a formulation that treats mobility as complementary supervision for learning POI representations and evaluate this directly on map-enrichment tasks. This problem setting has not previously been explored for either text embeddings or mobility-based POI representations, and we consider this an additional source of novelty. For this reason, we do not evaluate ME-POIs on mobility-prediction tasks, like next location prediction. Specifically, we believe that adding such tasks would conflate two different problem settings, rather than provide a meaningful evaluation.

---

### Official Review · Reviewer_7kE3 · 2025-10-25

**Soundness:** 2
**Presentation:** 3
**Contribution:** 2
**Rating:** 2
**Confidence:** 4

**Summary:**

This paper proposes ME-POIs, a framework that augments static, text-based representations of POIs with mobility-derived behavioral signals from large-scale human movement data. Instead of relying solely on static textual or visual descriptions of what a place is, ME-POIs models how a place is used, by encoding contextualized embeddings of visits that incorporate temporal and sequential context. Specifically, the method employs a Transformer-based visit sequence encoder, contrastive learning between visit and global POI embeddings, and a multi-scale distribution transfer mechanism that propagates temporal usage patterns from anchor POIs to sparsely visited ones. Experiments on several datasets demonstrate that the proposed method outperforms baselines across map-enrichment tasks such as predicting opening hours, permanent closures, popularity, and price levels.

**Strengths:**

S1. While most prior work has focused on static or semantic information of POIs, this study reframes the problem by emphasizing how places are actually used under human mobility rather than static features.

S2. The authors introduce a dynamic behavioral dimension that captures visitation frequency, timing, and sequential context, and consider the sparsity characteristics of POIs in the model design.

S3. Experiments show that the proposed method achieves the best performance across several POI-centric tasks.

**Weaknesses:**

W1. The novelty of this method is limited. The models applied to learn transition patterns, including location/time encoding, Transformer-based trajectory encoders and InfoNCE contrastive learning, are directly from prior works in mobility prediction or location representation learning. The framework mainly combines these existing ingredients rather than introducing new modeling mechanisms. Consequently, the contribution may be perceived as an incremental combination of established techniques.

W2. The design of the visit distribution transfer mechanism lacks conceptual clarity. The paper suggests transferring temporal visitation patterns from anchor POIs to sparse ones, but the rationale for establishing such connections is not well-justified. It remains unclear why similarity in spatial proximity or limited co-visitation patterns should imply that sparse POIs should inherit anchor-level visit distributions. A more rigorous explanation or empirical validation of this assumption would strengthen the argument.

W3.  In Line 288, the model introduces mixture weights learned individually for each sparse POI. This design raises a potential inconsistency: if sparse POIs are characterized by few visits and limited data, assigning additional learnable parameters to each of them does not seem to be sound. The approach seems to contradict the underlying motivation of handling sparsity, as the introduced parameters might overfit or fail to generalize given the scarcity of observations.

W4. For text alignment loss (Eq. 12), it aims to derive similar representations for its text embedding and $z^{ME}_{p}$. However, if this alignment objective is fully achieved, the two representations would converge to nearly identical spaces, potentially eliminating the complementary information that the mobility modality is meant to contribute. In other words, the text-alignment loss may unintentionally collapse the two modalities rather than encouraging mutual enrichment.

W5. The reproducibility of this work is problematic. The proposed framework assumes access to large-scale, high-quality mobility traces (e.g., Veraset datasets) that are proprietary and not publicly available, and the same limitation applies to the evaluation datasets. As a result, it would be difficult for the research community to reproduce or extend the experiments.

W6. The comparison between ME-POIs and text-embedding baselines may not be entirely fair. All baselines are pretrained general text models used off-the-shelf, while ME-POIs receive additional fine-tuning on mobility data. It is not surprising that the method outperforms these baselines.

**Questions:**

Q1. The inclusion of per-POI mixture weights (Line 288) seems counterintuitive under data sparsity, as each sparse POI is associated with limited visits.

Q2. The authors are suggested to further explain the alignment loss in Eq. (12), which may risk collapsing the two modalities into redundant representations.

Q3. Please refer to other comments in Weaknesses.

---

> ### Author Response · Authors · 2025-11-21
>
> We thank the reviewer for their comments. Please find answers to your questions below.
>
> **Comment W1:** *“The novelty of this method is limited.[...]”*
>
> **Response:** Prior methods that incorporate mobility signals for POIs primarily focus on **local, sequence-level transitions** (e.g., next-POI prediction), where Transformer approaches are used as analogues of next-word models. In contrast, our framework addresses a different and previously unexplored problem: **learning global, usage-aware POI representations** that capture how places are visited across many users, times, and contexts, and can serve as strong complementary embeddings to text-derived POI representations to address POI-centric inference tasks.
>
> Because this problem requires modeling aggregated behavioral patterns, our framework begins with the standard use of the Transformer encoder architecture for visit encoding but introduces two components not found in prior work: (i) a global contrastive learning objective that aligns each contextualized visit embedding with a shared, learnable POI embedding, to accumulate population-level behavioral information from all its visits; and (ii) a multiscale distribution transfer mechanism that provides temporal visitation priors for sparsely visited POIs. To our knowledge, neither of these components has been explored in prior work. In particular, we are not aware of any method that uses contrastive learning to aggregate long-term usage patterns (mobility-based or otherwise )into global POI embeddings. Likewise, no work has demonstrated that such mobility signals can meaningfully improve text-based POI representations derived from modern text encoders.
>
> We have additionally included new experiments  (please see Table 3&4 in Section 5) showing that POI embeddings derived from sequence-level mobility models used in next-location prediction tasks perform significantly worse on our static downstream tasks compared to embeddings built from our ME-POIs, which is expected given that the objectives of these next-location prediction models differ fundamentally from those of general-purpose POI representation learning.
>
> In line with the scope of ICLR, our work presents a **new representation-learning formulation for POIs**, and is the first modeling framework that implements this idea. We believe that this conceptual shift expands the field and creates space for future research to incorporate more sophisticated architectures within this formulation.
>
> **Comment W2:** *“The design of the visit distribution transfer mechanism lacks conceptual clarity[...]”*
>
> **Response:** The distribution-transfer module does not assume that sparse POIs should “inherit’’ an anchor’s pattern. Its role is to provide a smooth prior only when the empirical visit distribution is too sparse to estimate reliably. This follows two well-established principles: (i) human activity exhibits spatial locality. Nearby POIs are typically in the same time zone and share land-use and accessibility conditions that result in similar mobility patterns (e.g., office-area morning peaks, residential evening peaks) and (ii) POIs within the same micro-environment (e.g., a residential block or commercial street) participate in the same neighborhood activity cycle, experiencing similar flows of commuters or residents. We also experimented with alternative similarity spaces, such as semantic similarity between POIs, and found that geographic distance consistently yielded the strongest results, following Tobler's first law of geography (see reference “Miller, 2004), further supporting spatial locality as the most robust prior for temporal pattern transfer. The multi-scale, softly weighted mixture uses these spatial signals only as a weak prior, gently regularizing sparse POIs toward plausible temporal shapes without forcing them to match any specific anchor.
>
> **Comment W3:** *“if sparse POIs are characterized by few visits and limited data, assigning additional learnable parameters to each of them does not seem to be sound [...]”.*
>
> **Response:** We would like to clarify that the mixture weights for sparse POIs are not learned from their own limited visit data, they are optimized only through $\tilde{r}_{p_s}$​​, which is a multi-scale mixture of distributions from the nearby anchor POIs. Thus, these weights function as a small gating mechanism that adjusts the contribution of different spatial scales of the anchor prior. We will revise the text to make this distinction explicit.

---

> ### Author Response · Authors · 2025-11-21
>
> **Comment W4:** *“The text-alignment loss may unintentionally collapse the two modalities rather than encouraging mutual enrichment.”.*
>
> **Response:** We agree with the reviewer that, in principle, a text-alignment objective could collapse the two modalities if applied too strongly. In our framework, the text-alignment loss does not dominate or collapse the two modalities for two reasons. First, the alignment is applied with a small weight and only encourages partial consistency, while the mobility embedding is simultaneously optimized by three other objectives (contrastive visit alignment, anchor KL, sparse KL). The contrastive visit alignment is our primary optimization objective, ensuring that the embeddings retain mobility-aware structure. Second, the alignment is implemented via a learned linear projection of the text embedding into the mobility space, allowing the two representations to remain distinct while still being comparable for downstream tasks. Our robust empirical results support this claim; we observe that ME-POIs consistently improves over text-only baselines, indicating that the mobility information is preserved.
>
>
> **Comment W5:** *“The reproducibility of this work is problematic.”*
>
> **Response:** While the Veraset datasets are not public, they are widely used in academic work and available to researchers through a standard data-use request (similar to SafeGraph and other commercial geospatial datasets). Furthermore, all components of our framework are fully reproducible with any mobility dataset that provides standard GPS traces. We also provide complete code, implementation details, and hyperparameters, which will allow researchers to reproduce the methodology even if they use a different geographical area or dataset. Lastly, we plan to make our learned POI embeddings publicly available upon acceptance.
>
> **Comment W6:** *“The comparison between ME-POIs and text-embedding baselines may not be entirely fair. All baselines are pretrained general text models used off-the-shelf, while ME-POIs receive additional fine-tuning on mobility data. It is not surprising that the method outperforms these baselines.”*
>
> **Response:** Our goal in these experiments is not to argue that ME-POIs outperforms all text-based models, but to show the added value of incorporating mobility-derived representations on top of strong static POI embeddings.  We showed that even for tasks that appear static, mobility provides behavioral context, how POIs function in practice, that static information does not capture, which explains why it can yield measurable improvements.
>
> Furthermore, we would like to clarify that ME-POIs and all text-only baselines use the same fine-tuning pipeline: in every case, we freeze the pretrained encoder and train an identical lightweight MLP head per downstream task. The only additional step for ME-POIs is a self-supervised pretraining stage on mobility data to produce mobility-aware POI embeddings.
>
> We also emphasize that the downstream tasks (opening hours, permanent closure, popularity, and price level) are attributes that, in principle, are accessible to web-scale text models (e.g., via business descriptions, reviews, or aggregated statistics). The fact that mobility-only embeddings (without any POI text) already match or surpass several text baselines, and that combining mobility with text consistently yields the best performance, indicates that mobility provides complementary signals rather than simply benefiting from extra supervised tuning.

---

> > ### Comment · Reviewer_7kE3 · 2025-11-26
> >
> > Thanks for your response.
> >
> > I agree with the comments raised by Reviewer Fo4i. Conceptually, the new aspect is that you aggregate across all visits of a POI using contrastive learning so that a single embedding summarizes long-term usage (also with other modules). But this is essentially a design choice on what each embedding should represent (global vs. local), not a novel architecture or a modeling concept. I acknowledge that the task framing is useful in several downstream tasks, but the technical mechanisms are all standard and well-studied.
> >
> > Regarding the transfer module, the model still assigns a separate vector of mixture weights of dimension M to each sparse POI. These are POI-specific parameters that must be learned from gradients driven by the KL loss relative to the constructed prior. In long-tail regimes with many sparse POIs, this introduces many per-POI parameters that are only weakly constrained by a small amount of indirect supervision. This is exactly the kind of setting where overfitting or unstable estimates are likely. My concern is that the original motivation is to handle POIs with very few visits. However, the transfer module introduces more POI-specific parameters. Even if these weights are learned via the prior rather than raw counts, you still end up with a larger parameter surface tied to POIs that lack direct data.
> >
> > While I appreciate the clarifications and the additional experiments, I remain unconvinced that the overall concept and design are sufficiently novel. I will adjust the score in response to your efforts, and refer to other reviewers in the discussion phase.

---

> ### Author Response · Authors · 2025-11-29
>
> We respectfully disagree with the reviewer’s claim that our contribution is “not novel” because it does not introduce a new architecture. In representation learning, novelty is not only defined by adding new layers, but also by changing what is learned and how it is supervised. Our work explicitly changes the learning target from modeling user movement to learning place-level representations, which is a modeling contribution. If this were just a design choice, existing POI baselines would be expected to perform similarly on our downstream tasks (which are also newly introduced and not shown in previous works), which they do not. The large performance gap indicates that the learned embeddings capture fundamentally different information.
>
> Regarding the mixture of weights, we use M=3 which is a small number. We do not observe any overfitting or instability during training, and our ablation study (Table 5) shows that adding $L_{sparse}​$ improves downstream performance.

---

### Official Review · Reviewer_dq6D · 2025-10-30

**Soundness:** 3
**Presentation:** 3
**Contribution:** 3
**Rating:** 6
**Confidence:** 4

**Summary:**

- This paper introduces Mobility-Embedded POIs (ME-POIs), a novel framework for learning *behavior-aware* point-of-interest (POI) representations by embedding large-scale human mobility patterns into otherwise static POI embeddings. The core motivation is that conventional POI representations—derived from text, images, or spatial coordinates—capture *what a place is*, but fail to describe *how a place is used*. To address this gap, the authors model mobility sequences as temporal visit events and design a contrastive sequence encoding mechanism that aligns context-aware visit embeddings with global POI embeddings.

**Strengths:**

- Novel integration of human mobility and semantic signals
  - The paper convincingly bridges the gap between *what a place is* (semantic) and *how it is used* (behavioral), which has been largely overlooked in prior POI representation work.

- Effective solution to data sparsity via multi-scale distribution transfer
  - The spatially grounded, multi-scale transfer of visit-time distributions is conceptually sound and empirically effective.

- Comprehensive experimental validation
  - The paper uses two large, real-world mobility datasets and evaluates on diverse, practically relevant tasks. Performance improvements are substantial and consistent across baselines and metrics.

**Weaknesses:**

- **Incomplete and potentially outdated baseline comparisons**
  - The experimental design lacks a critical comparison against state-of-the-art methods that learn representations specifically for user mobility modeling. While the proposed POI representation is designed for POI-centric tasks (e.g., business hour prediction), the baselines are limited to text-based embeddings. This fails to demonstrate that the learned representation is superior to existing mobility-specific representations for tasks like next-POI prediction.
  - Furthermore, the paper does not adequately justify the recency and competitiveness of the chosen text-based baselines, raising concerns about whether the improvements are measured against a strong, contemporary benchmark.
- **Typos problem**
  - The expression in Equation (12) is incorrect. It is intended to project the text embedding into the mobility embedding space. However, the current expression incorrectly assigns the parameter W to the mobility embedding.
  - In Section 3 Problem Formulation, the definition of the POI attribute xi contains redundant set membership symbols (e.g., ∈∈R2).

**Questions:**

- The proposed multi-scale distribution transfer module is specifically designed to address the long-tail sparsity problem in POI visits. The experimental results do not provide a separate evaluation of model performance on dense (frequently visited) versus sparse (rarely visited) POIs. Could the authors clarify how ME-POIs performs across these two subsets?

- Could the authors explain why the integration with ME-POIs leads to a performance drop on the Gemini baseline, and whether similar issues occur with other text embedding models?

---

> ### Author Response · Authors · 2025-11-20
>
> **Comment W1:** *“The experimental design lacks a critical comparison against state-of-the-art methods that learn representations specifically for user mobility modeling.”*
>
> **Response:** We appreciate the reviewer's suggestion. In the revised manuscript, we have added a comprehensive comparison against mobility-based models used in next-location prediction. These include both (i) POI representation approaches that explicitly learn POI embedding vectors (Skip-Gram, POI2Vec, Geo-Teaser, TALE, HIER, CTLE) and (ii) sequence-prediction models that provide POI embeddings implicitly through a learnable token-embedding layer (DeepMove, STAN, Graph-Flashback, GETNext, and TrajGPT). These results are now included in Section 5 and Tables 3 and 4, and are highlighted in blue for clarity. Below we include the results for the Los Angeles dataset:
>
> | Method | Open Hours (F1/AUC) | Perm. Closure (F1/AUPRC) | Popularity (AUC/AUPRC) | Price (Acc/F1) |
> |-------|----------------------|----------------------------|--------------------------|----------------|
> | Skip-Gram | 0.462±0.002 / 0.520±0.006 | 0.649±0.008 / 0.123±0.004 | 0.530±0.001 / 0.268±0.001 | 0.564±0.007 / 0.286±0.004 |
> | POI2Vec | 0.460±0.003 / 0.482±0.004 | 0.564±0.039 / 0.112±0.005 | 0.519±0.003 / 0.263±0.002 | 0.530±0.014 / 0.249±0.013 |
> | Geo-Teaser | 0.460±0.002 / 0.470±0.005 | 0.448±0.083 / 0.116±0.004 | 0.523±0.007 / 0.266±0.003 | 0.511±0.009 / 0.194±0.023 |
> | TALE | 0.461±0.002 / 0.464±0.006 | 0.375±0.197 / 0.102±0.003 | 0.486±0.006 / 0.248±0.003 | 0.504±0.005 / 0.189±0.027 |
> | HIER | 0.473±0.002 / 0.547±0.004 | 0.660±0.005 / 0.119±0.001 | 0.569±0.005 / 0.291±0.001 | 0.529±0.029 / 0.229±0.047 |
> | CTLE | 0.463±0.001 / 0.511±0.007 | 0.115±0.102 / 0.098±0.006 | 0.501±0.006 / 0.249±0.003 | 0.488±0.015 / 0.244±0.008 |
> | DeepMove | 0.460±0.003 / 0.484±0.007 | 0.370±0.135 / 0.110±0.002 | 0.494±0.006 / 0.253±0.001 | 0.503±0.009 / 0.224±0.030 |
> | STAN | 0.464±0.002 / 0.509±0.007 | 0.220±0.215 / 0.099±0.007 | 0.550±0.006 / 0.250±0.002 | 0.497±0.012 / 0.248±0.006 |
> | Graph-Flashback | 0.463±0.002 / 0.506±0.008 | 0.233±0.203 / 0.099±0.007 | 0.504±0.007 / 0.251±0.002 | 0.496±0.017 / 0.248±0.009 |
> | GETNext | 0.431±0.007 / 0.500±0.001 | 0.200±0.220 / 0.103±0.004 | 0.503±0.001 / 0.252±0.005 | 0.410±0.092 / 0.220±0.032 |
> | TrajGPT | 0.483±0.003 / 0.491±0.005 | 0.215±0.120 / 0.101±0.006 | 0.496±0.006 / 0.249±0.003 | 0.475±0.015 / 0.237±0.009 |
> | *ME-POIs (w/o text-align)* | *_0.540±0.002_ / _0.703±0.005_* | *_0.757±0.025_ / _0.154±0.006_* | *_0.633±0.004_ / _0.337±0.005_* | *_0.600±0.011_ / _0.308±0.005_* |
> | **ME-POIs** | **0.554±0.004 / 0.722±0.005** | **0.766±0.023 / 0.161±0.005** | **0.653±0.004 / 0.355±0.008** | **0.609±0.018 / 0.322±0.012** |
> |-------|----------------------|----------------------------|--------------------------|----------------|
>
> We report the performance of ME-POIs both with and without the text-alignment objective (w/o $L_\text{text-align}$), given that the baselines do not use any text signal. Our evaluation shows that our mobility-only variant outperforms every mobility-based baseline across all tasks and in both cities, supporting our argument that next-location prediction models are not suitable for static, place-centric prediction tasks. The full ME-POIs model, which also incorporates the text-alignment objective, achieves the strongest overall performance.
>
> **Comment W2:** *“The paper does not adequately justify the recency and competitiveness of the chosen text-based baselines, raising concerns about whether the improvements are measured against a strong, contemporary benchmark.”*
>
> **Response:** To our knowledge, our baselines already include the most competitive and widely used text embedding models. Specifically, our baselines include MPNet, E5-large-v2, GTR-T5-large, Nomic-Embed-v1, OpenAI text-embedding-3-small/large, and Gemini embedding-001, covering both state-of-the-art academic models (2022-2024) and current industry-grade commercial embeddings. These are among the strongest text encoders available today and we believe they constitute a good benchmark for evaluating the added value of mobility-derived information. Beyond these baselines, we would be happy to incorporate any additional text embedding models that the reviewer thinks are appropriate.
>
> **Comment W3:** *“The experimental results do not provide a separate evaluation of model performance on dense (frequently visited) versus sparse (rarely visited) POIs.”*
>
> **Response:** We have now added the suggested experiment as a case study under Section 5. The added experiment is highlighted in blue in the updated manuscript. The results confirm that our proposed distribution transfer module improves representation quality for both rarely and frequently visited POIs.

---

> > ### Author Response · Authors · 2025-11-20
> >
> > **Comment W4:** *“Could the authors explain why the integration with ME-POIs leads to a performance drop on the Gemini baseline, and whether similar issues occur with other text embedding models?”*
> >
> > **Response:** This small performance drop occurs *only in F1* and *only* for Gemini on the Permanent Closure task. Because the task has highly imbalanced labels (Table 7), we believe this small drop might be due to minor shifts in predicted probabilities (common when adding a new embedding signal) that slightly change the optimal threshold and artificially lower F1. We note that our AUPRC improves by 2.21%, which highlights that the ranking quality actually improves. Overall, across all other baselines, tasks, and metrics, ME-POIs **consistently improves performance**, often by a substantial margin, as reported.

---

### Official Review · Reviewer_hFqi · 2025-10-31

**Soundness:** 3
**Presentation:** 3
**Contribution:** 2
**Rating:** 6
**Confidence:** 5

**Summary:**

To address the need for dynamic POI representation, the paper introduces Mobility Embedded POIs (ME-POIs)—a pretraining framework that enriches static text-based POI embeddings with mobility signals from visit sequences. It generates contextualized embeddings for each visit by integrating the POI’s static attributes with temporal and sequential context (e.g., visit timing, preceding/following visits). To tackle sparsely visited long-tail POIs, the framework transfers visit distributions from data-rich locations to sparse ones, utilizing multi-scale spatial proximity to capture local and regional patterns.

**Strengths:**

- Existing methods, while effective in predicting mobility behaviors, fail to be explicitly designed for and directly transferable to place-centric tasks that require an understanding of long-term, aggregated patterns of place usage and function. This article addresses this gap by introducing the Mobility-Embedded POIs (ME-POIs) framework.
- The proposed model augments static POI representations from text embedding models by directly integrating large-scale human mobility signals. Starting from visit sequences, each visit is encoded into a contextualized embedding that reflects the POI’s static attributes and its temporal context within mobility patterns. These visit-level embeddings are aligned with a learnable POI embedding via contrastive learning, ensuring each POI representation incorporates aggregated behavioral information over time and across users.
- For rarely visited POIs with data sparsity issues, a distribution transfer mechanism is proposed. It propagates temporal usage patterns from nearby, frequently visited POIs across multiple spatial scales to those with limited data. This multi-scale strategy captures local and regional behavioral trends, yielding high-quality POI embeddings even in the long tail of the visit distribution.

**Weaknesses:**

1. The experimental section includes relatively few baselines and does not separately analyze the impact of sparse POI distributions versus dense POI distributions on the results.
2. ME-POIs is proposed as a pre-trained POI representation learning framework, but the paper does not discuss its ability for cross-scenario generalization. It remains unclear whether the framework can be adapted to new POI scenarios via few-shot fine-tuning, or if it requires full re-pretraining.

**Questions:**

1. Please analyze the computational efficiency of the pre-training process in this framework.
2. How does the framework perform on datasets with a larger number of POIs, such as Gowalla, Foursquare, and Weeplaces?

---

> ### Author Response · Authors · 2025-11-20
>
> **Comment W1:** *“The experimental section includes relatively few baselines and does not separately analyze the impact of sparse POI distributions versus dense POI distributions on the results.”*
>
> **Response:** We appreciate the reviewer's suggestion. We have now added the suggested experiment as a case study under Section 5. The added experiment is highlighted in blue in the updated manuscript. The results confirm that our proposed distribution transfer module improves representation quality for both rarely and frequently visited POIs.
>
> **Comment W2:** *“[...] the paper does not discuss its ability for cross-scenario generalization. It remains unclear whether the framework can be adapted to new POI scenarios via few-shot fine-tuning, or if it requires full re-pretraining.”*
>
> **Response:** We assume that by "new POI scenarios", the reviewer is referring to cases of newly introduced (previously unseen) POIs. For such POIs, ME-POIs can naturally generate their representations through lightweight fine-tuning of only the new POI’s embedding vector, as long as a minimal mobility signal is available. Because the visit encoder operates on continuous spatiotemporal features (not POI IDs), it learns general mobility patterns and can already produce meaningful contextualized embeddings for visits to a newly introduced POI (this is similar to the case of a sparsely visited POI). Our multiscale kernel module further provides a visitation-pattern prior from nearby anchor POIs, stabilizing the embedding under sparse observations. As a result, new POIs can be integrated without retraining the full model. If the reviewer intended something else by “new POI scenarios”, we would appreciate clarification and would be happy to address it.
>
> **Comment W3:** *“Please analyze the computational efficiency of the pre-training process in this framework.”*
>
> **Response:** The pre-training cost of ME-POIs is primarily driven by running the visit encoder on sequences of visits. For a sequence length of $L$ and an embedding dimension $d$, the overall computation complexity is $O(L^2 \cdot d + L \cdot d^2)$. The contrastive module operates only over in-batch negatives: for a batch of $B$ visits containing $U$ unique POIs, its cost is $O(B \cdot U \cdot d)$, which in practice remains lightweight and independent of the full POI set size. Note that the \# of unique POIs in the batch is less than or equal to \# of visits in the batch. The POI anchor distributions and multiscale kernels are precomputed only once offline, with computation complexity $O(M \cdot |P_{anchor}| \cdot |P_{sparse}| )$ for $M$ scales. In practice, our model is lightweight with ~53.7 M parameters, well within standard computational budgets. We have added this explanation to the Appendix of the revised manuscript.
>
> **Comment W4:** *“How does the framework perform on datasets with a larger number of POIs, such as Gowalla, Foursquare, and Weeplaces?”*
>
> **Response:** While the public datasets cited above include a large number of POIs, the visit patterns are extremely sparse, which means they do not meaningfully stress the scalability of our model. As explained in our previous response, the pre-training complexity of ME-POIs is driven by the volume of visits (not by the size of the POI vocabulary), since both the encoder and the contrastive module operate strictly on visit sequences and in-batch POI negatives. Thus, ME-POIs would remain computationally efficient even on these public datasets. In contrast, the Veraset dataset provides substantially denser and more challenging mobility signals, making it a far more realistic benchmark for evaluating the scalability and effectiveness of our approach.

---

### Meta-Review · Area_Chair_6cv7 · 2026-01-07

**Summary:**

This paper proposes Mobility-Embedded POIs (ME-POIs), a framework for learning POI representations by integrating large-scale human mobility signals with static text-based embeddings. Reviewers agree that the motivation is clear and relevant, and that incorporating behavioral usage into POI representations is a meaningful direction. The method demonstrates empirical improvements on several map-enrichment tasks. However, multiple reviewers raise concerns about limited technical novelty, conceptual justification of the distribution transfer mechanism, and the generality and reproducibility of the proposed approach.

**Reviewer Concerns:**

The authors addressed several concrete issues by adding baseline comparisons, sparse-versus-dense analyses, and computational cost explanations, but major concerns remain. Multiple reviewers remain unconvinced that the proposed framework introduces sufficiently novel modeling ideas beyond a recombination of established components such as Transformer encoders, contrastive learning, and alignment losses. The multi-scale distribution transfer for sparse POIs is still viewed as weakly justified, particularly due to the introduction of per-POI mixture weights under data sparsity. Concerns also persist about evaluation scope, fairness of comparisons to text-only baselines, and limited reproducibility due to reliance on proprietary mobility data. Overall, these issues are only partially addressed by the rebuttal.

**Reviewer Scores:**

Reviewer hFqi would likely remain 6. Reviewer dq6D would likely remain 6. Reviewer 7kE3 mentioned "I remain unconvinced that the overall concept and design are sufficiently novel. I will adjust the score in response to your efforts," and would likely remain at 2 with the possibility of increasing to 4, taking the further explanation from the author for consideration. Reviewer Fo4i would likely remain at 2, continuing to view the contribution as incremental and insufficiently distinct from prior work.

---

### Decision · Program_Chairs · 2026-01-26

Reject